# Predicting postoperative peritoneal metastasis in gastric cancer with serosal invasion using a collagen nomogram

Dexin Chen [1,2,10], Zhangyuanzhu Liu[3,10], Wenju Liu[1,4,5,10], Meiting Fu [6,10], Wei Jiang[1], Shuoyu Xu[1,7], Guangxing Wang [2,8], Feng Chen[9], Jianping Lu[4,5], Hao Chen[1], Xiaoyu Dong[1], Guoxin Li [1✉], Gang Chen [4,5✉], Shuangmu Zhuo [2✉] & Jun Yan [1,5✉]

Accurate prediction of peritoneal metastasis for gastric cancer (GC) with serosal invasion is crucial in clinic. The presence of collagen in the tumour microenvironment affects the metastasis of cancer cells. Herein, we propose a collagen signature, which is composed of multiple collagen features in the tumour microenvironment of the serosa derived from multiphoton imaging, to describe the extent of collagen alterations. We find that a high collagen signature is significantly associated with a high risk of peritoneal metastasis ($P < 0.001$). A competing-risk nomogram including the collagen signature, tumour size, tumour differentiation status and lymph node metastasis is constructed. The nomogram demonstrates satisfactory discrimination and calibration. Thus, the collagen signature in the tumour microenvironment of the gastric serosa is associated with peritoneal metastasis in GC with serosal invasion, and the nomogram can be conveniently used to individually predict the risk of peritoneal metastasis in GC with serosal invasion after radical surgery.

[1] Department of General Surgery, Nanfang Hospital, The First School of Clinical Medicine, Southern Medical University, Guangzhou 510515, China. [2] School of Science, Jimei University, Xiamen 361021 Fujian, China. [3] Department of Hepatobiliary and Pancreatic Surgery, Guangdong Provincial Hospital of Traditional Chinese Medicine, The Second Affiliated Hospital of Guangzhou University of Traditional Chinese Medicine, Guangzhou 510120, China. [4] Department of Pathology, Fujian Medical University Cancer Hospital and Fujian Cancer Hospital, Fuzhou 350014, China. [5] Precision Medicine Center, Fujian Provincial Cancer Hospital, Fuzhou 350014, China. [6] Department of Gastroenterology, Nanfang Hospital, Southern Medical University, Guangzhou 510515, China. [7] Department of Radiology, Sun Yat-sen University Cancer Center, Guangzhou 510060, China. [8] Key Laboratory of OptoElectronic Science and Technology for Medicine of Ministry of Education, Fujian Normal University, Fuzhou 350007, China. [9] Department of Oncological Surgery, The Second Affiliated Hospital of Fujian Medical University, Quanzhou 362000, China. [10] These authors contributed equally: Dexin Chen, Zhangyuanzhu Liu, Wenju Liu, Meiting Fu. ✉email: gzliguoxin@163.com; naichengang@126.com; shuangmuzhuo@gmail.com; yanjunfudan@163.com

Gastric cancer (GC) is one of the most commonly diagnosed malignant diseases and the second leading cause of cancer-related deaths worldwide[1]. The peritoneum is the most frequent metastatic site for GC after radical surgery[2,3]. The median survival time is only 4 months once peritoneal metastasis is diagnosed, compared with 14 months in GC without peritoneal metastasis[2,3]. Serosal invasion is the strongest indicator of peritoneal metastasis in GC[4,5]. Most patients with GC with serosal invasion will succumb to peritoneal metastasis within 2 years after surgery despite radical gastrectomy[6]. Therefore, early detection of peritoneal metastasis is integral to improving the prognosis of GC patients with serosal invasion.

Currently, there are two therapies for preventing peritoneal metastasis for GC patients with serosal invasion: extensive intraoperative peritoneal lavage (EIPL) and intraperitoneal chemotherapy (IPC)[7]. To date, the safety and efficacy of EIPL have been proven, but the long-term oncological outcomes are still unclear[8]. IPC was used to eliminate suspected malignant cells through locoregional chemotherapy. Several studies have reported that IPC is favourable for improving the oncological outcome and decreasing an incidence of peritoneal metastasis in GC with serosal invasion[9,10]. However, a considerable number of GC patients will not suffer from peritoneal metastasis despite serosal invasion. Moreover, IPC is costly and associated with an increased rate of postoperative complications, including digestive fistula, haematologic toxicity and systemic sepsis[11]. Thus, accurate prediction of the risk of peritoneal metastasis after radical gastrectomy is extremely important for the choice of IPC in GC with serosal invasion.

Peritoneal metastasis is difficult to predict on clinical grounds. Cytologic examination of peritoneal lavage, which has been used to assess the risk of peritoneal metastasis in GC with serosal invasion, has been reported to lack sensitivity because a large number of patients still die from peritoneal metastasis even though they have negative cytologic results[12]. Some imaging modalities, including computed tomography (CT) and endoscopic ultrasonography (EUS), are common examination tools for GC; however, the accuracy of these imaging modalities for the diagnosis of peritoneal metastasis is not satisfactory[13], and it is not until patients are suffering from peritoneal metastasis that these imaging modalities can identify the outcome. Considering the limited performance of the clinical variables and the high complication rates of IPC, a novel biomarker is needed for the prediction of peritoneal metastasis in GC with serosal invasion after radical gastrectomy to influence decision making.

It has been revealed that collagen alterations in the tumour microenvironment are correlated with cancer dissemination and prognosis[14–17]. The increased collagen density around cancer cells directs local invasion and metastasis[14]. Moreover, the radial alignment of collagen at the tumour-stroma boundary improves the invasiveness of cancer cells[15–17]. Previous investigations have shown that serosal changes predict peritoneal metastasis in GC with serosal invasion[18,19]. As the main component of the serosa, collagen can be quantified to determine the serosal changes in GC[20]. Thus, we hypothesize that collagen alterations in the tumour microenvironment of the serosa are associated with peritoneal metastasis in GC with serosal invasion.

Over the past decade, multiphoton imaging has emerged as a powerful tool for visualizing the assembly of collagen in tissues at a supramolecular level because of its underlying physical origin[21]. Multiphoton imaging is sensitive to changes in collagen and provides multiple quantitative metrics, including morphological and textural features, for diagnosing and predicting diseases[22,23]. Here, we propose a collagen signature with high-throughput quantitative collagen features that are automatically extracted using multiphoton imaging, to comprehensively quantify the extent of collagen alterations in the tumour microenvironment.

The integration of multiple biomarkers into a single signature has the potential to substantially improve predictive value over that of a single biomarker[24,25]. The least absolute shrinkage and selection operator (LASSO) regression is an effective approach for the regression of high-dimensional parameters and has been broadly applied for prognostic analysis[24–26].

Therefore, in this study, we use multiphoton imaging and LASSO regression to establish a multi-feature-based classifier, i.e., a collagen signature, to predict peritoneal metastasis. For clinical use, we develop and validate a competing-risk nomogram that integrates the collagen signature and clinicopathological risk factors for the individual postoperative prediction of peritoneal metastasis in GC with serosal invasion.

## Results

**Participants**. The clinicopathological characteristics of the training and validation cohorts are summarized in Table 1. Of the 198 patients in the training cohort, the median age [interquartile range (IQR)] was 57 (47.75–63.25) years, with 137 (69.2%) men. Among the 145 patients in the validation cohort, the median age (IQR) was 57 (52–64) years, with 98 (67.6%) men. There was no significant difference between the training and validation cohorts (Supplementary Data 1).

In the training cohort, the median follow-up duration (IQR) was 37 (20.75–44) months. The 3-year overall survival (OS) and disease-free survival (DFS) rates were 60.1% and 49.5% (Supplementary Fig. 1a, b), respectively, and the median time

**Table 1 Characteristics of the patients in the training and validation cohorts.**

| Variable | Training cohort (n = 198) | Validation cohort (n = 145) | P value |
|---|---|---|---|
| **BMI, no. (%)** | | | |
| ≥24 kg/m² | 47 (23.7) | 37 (25.5) | 0.71 |
| <24 kg/m² | 151 (76.3) | 108 (74.5) | |
| **CEA, no. (%)** | | | |
| Elevated | 58 (29.3) | 39 (26.9) | 0.63 |
| Normal | 140 (70.7) | 106 (73.1) | |
| **CA 19-9, no. (%)** | | | |
| Elevated | 48 (24.2) | 29 (20.0) | 0.35 |
| Normal | 150 (75.8) | 116 (80.0) | |
| **Tumour location, no. (%)** | | | |
| Cardia of the stomach | 41 (20.7) | 44 (30.3) | 0.10 |
| Body of the stomach | 50 (25.3) | 36 (24.8) | |
| Antrum of the stomach | 107 (54.0) | 65 (44.9) | |
| **Tumour size, no. (%)** | | | |
| ≥4 cm | 114 (57.6) | 93 (64.1) | 0.22 |
| <4 cm | 84 (42.4) | 52 (35.9) | |
| **Lauren classification, no. (%)** | | | |
| Intestinal | 71 (35.9) | 61 (42.1) | 0.24 |
| Diffuse or mixed | 127 (64.1) | 84 (57.9) | |
| **Differentiation status, no. (%)** | | | |
| Well | 11 (5.6) | 6 (4.1) | 0.33 |
| Moderate | 39 (19.7) | 40 (27.6) | |
| Poor | 95 (48.0) | 60 (41.4) | |
| Undifferentiated | 53 (26.7) | 39 (26.9) | |
| **Lymph node metastasis, no. (%)** | | | |
| N0 | 52 (26.3) | 29 (20.0) | 0.13 |
| N1 | 45 (22.7) | 25 (17.2) | |
| N2 | 35 (17.7) | 40 (27.6) | |
| N3a | 38 (19.2) | 33 (22.8) | |
| N3b | 28 (14.1) | 18 (12.4) | |
| **Chemotherapy, no. (%)** | | | |
| Yes | 144 (72.7) | 93 (64.1) | 0.09 |
| No | 54 (27.3) | 52 (35.9) | |
| **Collagen signature, median (IQR)** | 0.047 (−0.247 to 0.397) | 0.056 (−0.031 to 0.187) | 0.43 |

The comparison of collagen signature between two cohorts is performed using a two-sided Mann–Whitney U test, and the rest variables are compared using a two-sided $\chi^2$ or Fisher's exact test.
*BMI* body mass index, *CA* carbohydrate antigen, *CEA* carcinoembryonic antigen, *IQR* interquartile range.

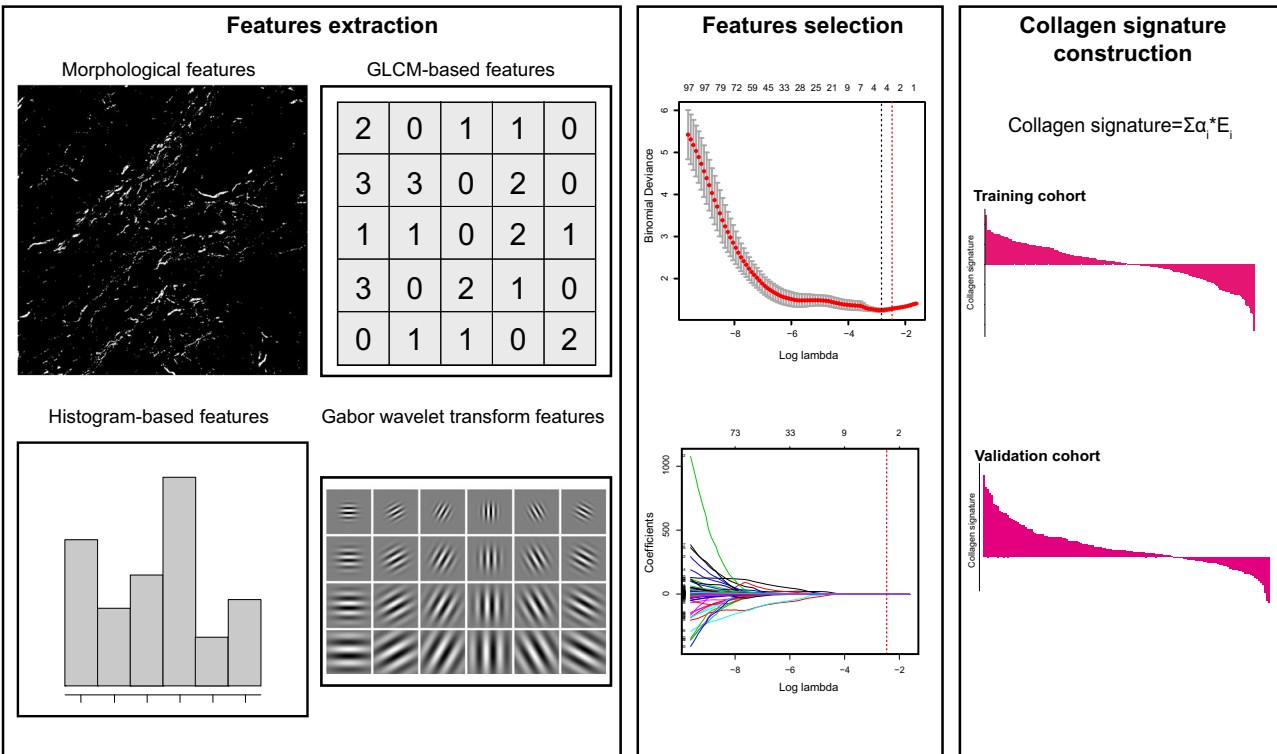

**Fig. 1 Construction framework of the collagen signature. a** Selection of the region of interest by comparing H&E staining and multiphoton imaging. The samples of all enroled 343 patients are used, and five regions of interest with a field of view of 500 × 500 μm per sample within the invasive region of the gastric serosa are randomly selected for multiphoton imaging. Scale bars: 1500 μm, 150 μm, 50 μm and 50 μm, respectively. **b** A total of 146 collagen features for each enroled patient are extracted from multiphoton images, and predictive collagen features are selected using LASSO regression in the training cohort, and the collagen signature is constructed. The collagen signature of the validation cohort is calculated from the collagen signature calculation formula obtained in the training cohort. GLCM, grey-level co-occurrence matrix; H&E, hematoxylin and eosin; LASSO, least absolute shrinkage and selection operator.

(IQR) to peritoneal metastasis was 14 (7–23) months. There were 79 (39.9%) patients with peritoneal metastasis (Supplementary Fig. 2), with 21 (10.6%) competing events (Supplementary Table 1). In the validation cohort, the median follow-up duration (IQR) was 50 (18.5–87.5) months, and the 3-year OS and DFS rates were 54.5% and 52.4% (Supplementary Fig. 1c, d), respectively. The median time (IQR) to peritoneal metastasis was 16 (10–26) months. The 3-year cumulative peritoneal metastasis rate was 29.7% (43/145) (Supplementary Fig. 2), with 26 (17.9%) competing events. The 122 patients who presented with peritoneal metastasis within 3 years after radical surgery either before or at the same time as recurrence at another site were the subjects in this analysis.

**Collagen signature establishment.** Four potential predictors from 146 collagen features were selected using LASSO regression (Fig. 1, Supplementary Fig. 3). The calculation formula for the collagen signature is presented in Supplementary Note 1, and the distribution of the collagen signature in the training and validation cohorts is listed in Supplementary Fig. 4. There was no significant difference in the collagen signature [median (IQR)] between the training (0.047, −0.247 to 0.397) and validation cohorts (0.056, −0.031 to 0.187) [median difference: −0.028; 95% confidence interval (CI): −0.100 to 0.042; $P = 0.43$].

**Assessment of the collagen signature with peritoneal metastasis and prognosis.** The optimal cutoff value generated by X-tile was

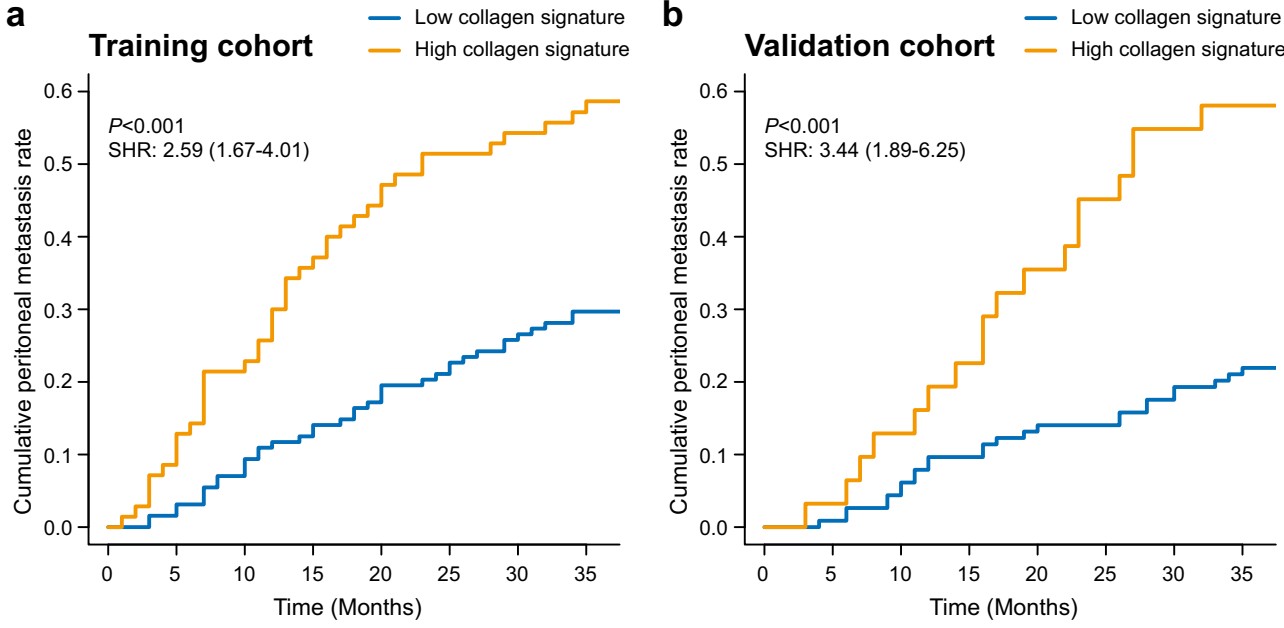

**Fig. 2 Collagen signature and peritoneal metastasis in the training and validation cohorts.** Cumulative peritoneal metastasis rate stratified by the collagen signature in the (**a**) training and (**b**) validation cohorts. The comparisons of the cumulative peritoneal metastasis between two subgroups are performed using a two-sided Gray's test. SHR, subdistribution hazard ratio.

0.2 (Supplementary Fig. 5) in the training cohort, and all 343 patients were classified into high and low collagen signature groups. The distribution of the clinical characteristics according to the high and low collagen signature groups in the training and validation cohorts is presented in Supplementary Table 2.

In the training cohort, there was a significantly higher 3-year cumulative peritoneal metastasis rate in patients with the high collagen signature (58.57% vs. 29.69%) than in those with the low collagen signature [subdistribution hazard ratio (SHR): 2.59; 95% CI: 1.67–4.01; $P < 0.001$] (Fig. 2a). Kaplan–Meier analysis showed that patients with the high collagen signature experienced a significantly shorter 3-year OS (39.8% vs. 71.1%; log-rank $P < 0.001$) and DFS (24.2% vs. 63.3%; log-rank $P < 0.001$) (Fig. 3a, b) than patients with the low collagen signature, with a hazard ratio (HR) of 2.27 (95% CI: 1.78–4.31; $P < 0.001$) and 3.09 (95% CI: 2.08–4.59; $P < 0.001$) for the 3-year OS and DFS rates, respectively, in the Cox regression analysis.

The same analyses were performed in the validation cohort. Similar results were found between the high and low collagen signature groups for the 3-year cumulative peritoneal metastasis rate (58.06% vs. 21.92%) (SHR: 3.44; 95% CI: 1.89–6.25; $P < 0.001$) (Fig. 2b). The 3-year OS rates of patients with the high and low collagen signatures were 25.8% and 63.2% (log-rank $P < 0.001$), respectively, along with 3-year DFS rates of 22.6% and 60.5% (log-rank $P < 0.001$) (Fig. 3c, d), respectively; the corresponding HRs were 2.60 (95% CI: 1.57–4.28; $P = 0.012$) and 2.66 (95% CI: 1.61–4.38; $P = 0.002$) for the 3-year OS and DFS rates, respectively.

**Development of the competing-risk nomogram.** As shown in Table 2, in the univariate analysis, the collagen signature, tumour size ≥ 4 cm, tumour differentiation status and lymph node metastasis were significantly associated with peritoneal metastasis (Supplementary Data 2). These factors were incorporated into the multivariate analysis, and a competing-risk nomogram was constructed based on the four factors (Fig. 4a).

**Performance evaluation and validation of the nomogram.** The time-dependent receiver operating characteristic (ROC) curve of the nomogram to predict peritoneal metastasis at 3 years in the training cohort is presented in Supplementary Fig. 6a, with an area under the receiver operating characteristic curve (AUROC) of 0.825 (95% CI: 0.765–0.885). The nomogram yielded an averaged concordance index (C-index) of 0.792 (95% CI: 0.784–0.798). The calibration curve showed good agreement among the estimations with the nomogram and actual observations (Fig. 4b). In the validation cohort, the AUROC at 3 years was 0.776 (95% CI: 0.699–0.853) (Supplementary Fig. 6b). Furthermore, the average C-index for the nomogram was 0.708 (95% CI: 0.692–0.726), and favourable calibration was also confirmed (Fig. 4c).

**Clinical usefulness.** Decision curve analysis revealed that if the threshold probability in the clinical decision was less than 67%, using the competing-risk nomogram to predict peritoneal metastasis would add more net benefit than the treat-all scheme and the treat-none scheme (Supplementary Fig. 7), which indicated that the competing-risk nomogram is clinically useful.

The maximum Youden index of 0.3913 of the ROC curve of the nomogram was selected as the optimal cutoff value in the training cohort, and patients were divided into high-risk and low-risk groups. We found that the sensitivity, specificity, accuracy, negative predictive value (NPV) and positive predictive value (PPV) of the nomogram in the training cohort were 82.3%, 82.4%, 82.3%, 87.7% and 75.6%, respectively. In the validation cohort, the sensitivity was 81.4%, the specificity was 60.8%, the accuracy was 66.9%, the NPV was 88.9%, and the PPV was 46.8%. In the total cohort, the sensitivity was 82.0%, the specificity was 72.4%, the accuracy was 75.8%, the NPV was 88.0%, and the PPV was 62.1% (Supplementary Table 3).

**Comparison with the clinicopathological model.** To evaluate the superiority of the nomogram based on the collagen signature over other easily obtained clinical variables, we excluded

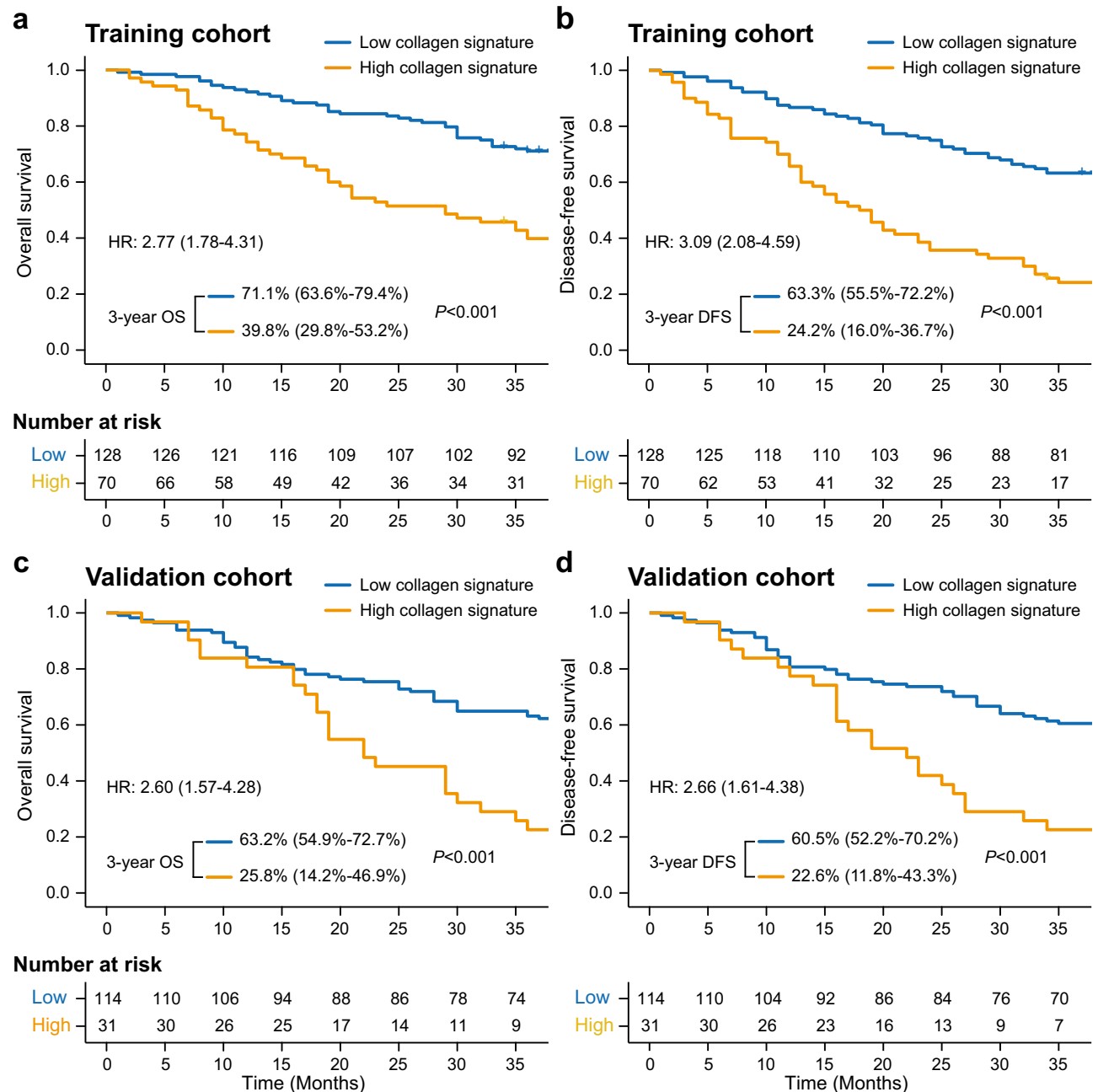

**Fig. 3 Kaplan–Meier survival analysis of the training and validation cohorts grouped by the collagen signature. a** Three-year OS comparison between the high and low collagen signatures in the training cohort. **b** The 3-year DFS comparison between the high and low collagen signatures in the training cohort. **c** The 3-year OS comparison between the high and low collagen signatures in the validation cohort. **d** The 3-year DFS comparison between the high and low collagen signatures in the validation cohort. The comparisons of OS and DFS between two subgroups are performed using a two-sided log-rank test. OS, overall survival; DFS, disease-free survival; HR, hazard ratio.

the collagen signature and built a clinicopathological model based on tumour size, tumour differentiation status and lymph node metastasis (Supplementary Table 4). The clinicopathological model yielded average C-indexes of 0.757 (95% CI: 0.748–0.765) and 0.676 (95% CI: 0.662–0.697) in the training and validation cohorts, respectively, and the nomogram based on the collagen signature presented a more robust ability to predict peritoneal metastasis in all enrolled patients [C-index comparison: 0.779 (95% CI: 0.773–0.786) vs. 0.736 (95% CI: 0.725–0.746), $P < 0.001$; $P < 0.001$ and $P = 0.016$ for the training and validation cohorts, respectively]

(Supplementary Table 5, Supplementary Data 3). Moreover, the 3-year AUROC of the clinicopathological model was 0.787 (95% CI: 0.724–0.850) and 0.721 (95% CI: 0.635–0.807) in the training and validation cohorts, respectively (Supplementary Fig. 8). Compared with the clinicopathological model, the nomogram based on the collagen signature also showed a significantly improved AUROC in all patients [AUROC comparison: 0.807 (95% CI: 0.760–0.855) vs. 0.762 (95% CI: 0.712–0.813), $P < 0.001$; $P = 0.01$ and 0.004 for the training and validation cohorts, respectively] (Supplementary Fig. 9 and Supplementary Table 6).

**Table 2 Univariate and multivariate Fine-Gray regression in the training cohort.**

| Variable | Univariate analysis | | Multivariate analysis | |
|---|---|---|---|---|
| | SHR (95% CI) | P value | SHR (95% CI) | P value |
| **Collagen signature** | 4.11 (2.58–6.56) | <0.001 | 2.49 (1.52–4.08) | <0.001 |
| **Age** | 0.99 (0.98–1.02) | 0.62 | | |
| **Sex** (Female vs. Male) | 0.87 (0.53–1.43) | 0.58 | | |
| **BMI** ($\geq$24 kg/m$^2$ vs. <24 kg/m$^2$) | 1.14 (0.69–1.89) | 0.61 | | |
| **CEA** (Elevated vs. Normal) | 1.08 (0.67–1.73) | 0.76 | | |
| **CA 19-9** (Elevated vs. Normal) | 1.58 (0.98–2.55) | 0.064 | | |
| **Tumour location** | | 0.56 | | |
| Antrum of the stomach | Reference | >0.99 | | |
| Body of the stomach | 1.32 (0.80–2.19) | 0.28 | | |
| Cardia of the stomach | 1.07 (0.60–1.90) | 0.82 | | |
| **Tumour size** ($\geq$4 cm vs. <4 cm) | 3.32 (1.99–5.52) | <0.001 | 2.35 (1.37–4.02) | 0.002 |
| **Lauren classification** (Diffuse or mixed vs. Intestinal) | 1.30 (0.82–2.07) | 0.27 | | |
| **Differentiation status** | | 0.001 | | 0.12 |
| Well + Moderate | Reference | >0.99 | Reference | >0.99 |
| Poor | 2.35 (1.18–4.69) | 0.015 | 1.42 (0.66–3.05) | 0.37 |
| Undifferentiated | 3.67 (1.81–7.47) | <0.001 | 2.12 (0.96–4.72) | 0.065 |
| **Lymph node metastasis** | | <0.001 | | 0.060 |
| N0 | Reference | >0.99 | Reference | >0.99 |
| N1 | 3.16 (1.35–7.44) | 0.008 | 2.47 (1.02–6.20) | 0.044 |
| N2 | 3.32 (1.33–8.24) | 0.01 | 2.50 (1.01–6.20) | 0.047 |
| N3a | 6.18 (2.67–14.29) | <0.001 | 3.22 (1.25–8.31) | 0.016 |
| N3b | 9.28 (3.94–21.87) | <0.001 | 4.43 (1.65–11.94) | 0.003 |
| **Chemotherapy** (Yes vs. No) | 0.75 (0.46–1.23) | 0.26 | | |

Association of all variables with peritoneal metastasis is analyzed using a two-sided Gray's test.
BMI body mass index, CA carbohydrate antigen, CEA carcinoembryonic antigen, CI confidence interval, SHR subdistribution hazard ratio.

## Discussion

An accurate assessment of peritoneal metastasis after radical gastrectomy is vital for decision making and improvement of prognosis in GC with serosal invasion. In this study, we found that the collagen signature in the serosal tumour microenvironment of GC with serosal invasion, which was constructed after multiphoton imaging, was significantly associated with peritoneal metastasis after radical surgery, and a competing-risk nomogram could predict peritoneal metastasis well, with satisfactory discrimination and calibration.

Compared to the clinicopathological model including tumour size, tumour differentiation status and lymph node metastasis, significant improvement in the C-index and AUROC was observed in the nomogram based on the collagen signature, which indicated that the collagen signature could improve the prediction of peritoneal metastasis beyond the use of easily obtained clinical variables.

Previous studies have demonstrated that peritoneal metastasis is caused by serosal invasion of the primary tumour and the subsequent shedding of malignant cells into the peritoneal cavity[5,27]. The magnitude of serosal changes is related to peritoneal metastasis[18,19]. Sun et al.[28] proposed that the extent of serosal invasion could be classified as the reactive type, nodular type, tendonoid type, and colour-diffused type according to the macroscopic serosal appearance. However, the determination of macroscopic serosal changes is subjective and qualitative and might vary from surgeon to surgeon[28]. Therefore, an objective and fully quantitative biomarker of the serosa is needed for the accurate prediction of peritoneal metastasis in GC with serosal invasion.

Currently, a diagnosis of peritoneal metastasis after radical surgery mainly depends on clinical signs, imaging examinations and even reoperation during the follow-up period; a practical prediction model at the time point of radical surgery to predict peritoneal metastasis in GC patients with serosal invasion is still lacking. In this study, although the peritoneal metastasis rate was considerable even in the low collagen signature group, a significantly higher peritoneal metastasis rate was found in the high collagen signature group, which indicates that the collagen signature could identify patients who were more likely to suffer from peritoneal metastasis after radical surgery. In addition, the nomogram yielded an overall sensitivity, specificity and accuracy of 82.0%, 72.4% and 75.8%, respectively, which are adequate for reassuring clinicians when selecting an appropriate population for interventions.

As the scaffold of the extracellular matrix, collagen accounts for most of its functions[29]. Our previous research revealed that collagen alterations in the tumour microenvironment of early GC significantly predicted lymph node metastasis[30]. Although peritoneal metastasis in GC with serosal invasion and lymph node metastasis in early GC indicate different metastatic procedures at different stages of GC, there are still certain changes in the extracellular matrix during disease progression[31]. From the calculation formula for the collagen signature, we found that the collagen signature was positively corrected with the cross-link density of collagen. In this study, the cross-link density indicates the connections between individual collagen fibres (i.e. physical cross-link density). A previous study has reported that an increased chemical cross-link density of collagen heightened the stromal stiffness and stimulated the invasive properties of tumour cells[32]. Thus, whether there is any connection between the physical cross-link density and chemical cross-link density and how the physical cross-link density affects the biological behaviours of tumour cells needs to be further investigated.

In this study, the collagen signature was constructed based on multiphoton imaging. Currently, with the development of interdisciplinary medicine, multiphoton imaging has been applied in the field of biomedical research[33,34]. It took only approximately 10

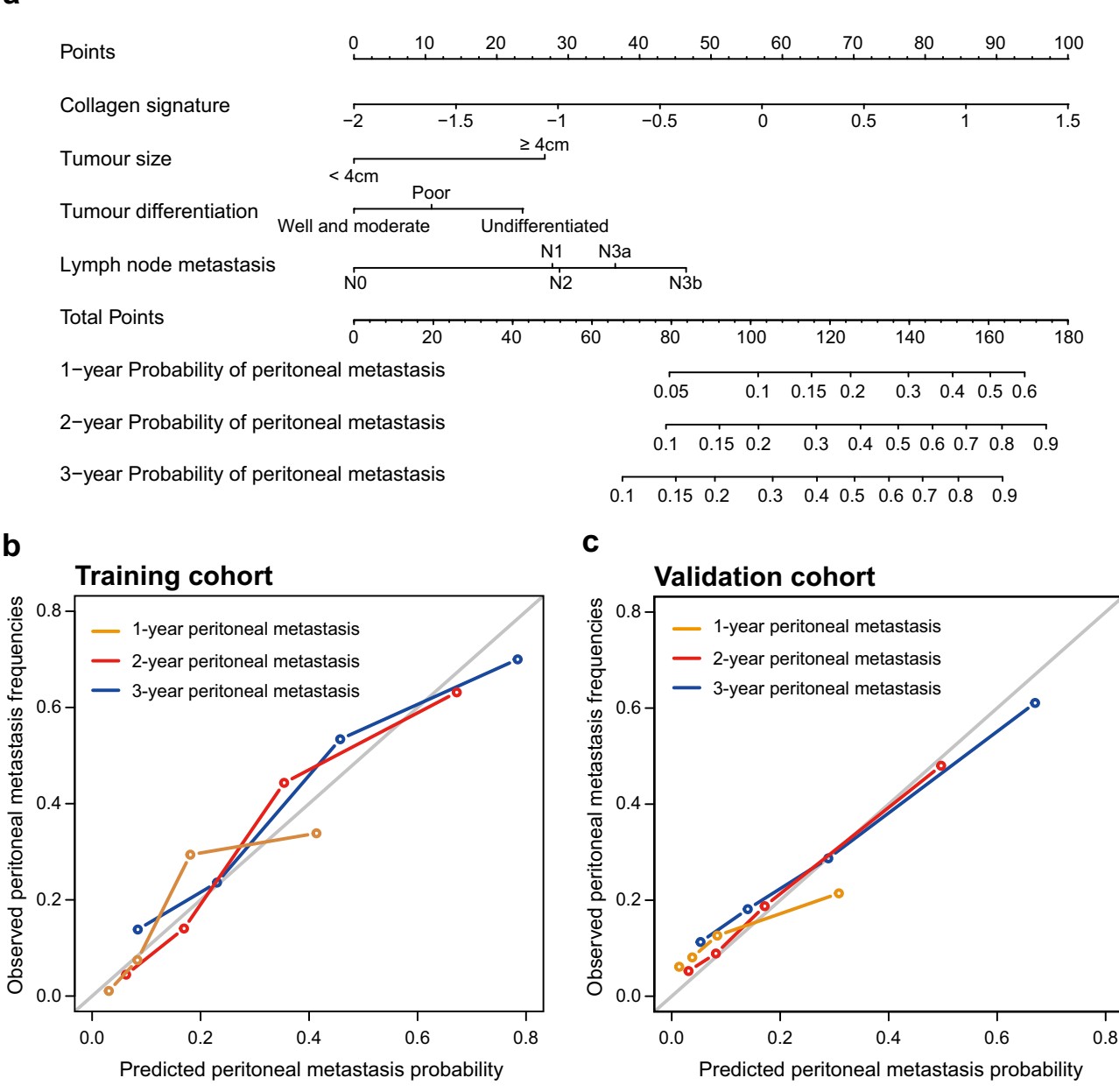

**Fig. 4 Competing-risk nomogram and the corresponding calibration curve. a** Competing-risk nomogram incorporating the collagen signature, tumour size, tumour differentiation and lymph node metastasis. **b** Calibration curve of the competing-risk nomogram in the training cohort. **c** Calibration curve of the competing-risk nomogram in the validation cohort.

min to complete the multiphoton imaging. There was no treatment on the unstained serial sections before the measurements, and the paraffin did not need to be removed[17,22]. Moreover, multiphoton imaging is a label-free and noninvasive tool to obtain the tissue structure and cell morphology of specimens; it is comparable to hematoxylin-eosin (H&E) staining and does not affect the collagen signature[35,36]; thus, experienced pathologists could master multiphoton imaging with little training, and it is possible to define regions of interest based on multiphoton imaging. In addition, it has been reported that tissue fixation and paraffin embedding have negligible effects on collagen detection and quantification; thus, a sample that was fixed overnight compared to one fixed over a few days, prior to paraffin embedding, would not influence multiphoton imaging[37]. Therefore, multiphoton imaging is promising for clinical transplantation.

Multiphoton imaging can visualize biomolecular arrays in cells, tissues and organisms; thus, the structural or molecular assignment may be linked to different collagen features. In this work, the high-throughput quantitative collagen features obtained suggest that the high-dimensional features of collagen including morphological and textural features from multiphoton imaging could be extracted after image processing. To date, a common consensus about the selection of collagen feature types has not yet been achieved to comprehensively quantify collagen alterations based on multiphoton imaging. The morphological features, such as collagen length and width, are easily understood. Histogram- and grey-level co-occurrence matrix (GLCM)-based features are two main types of textural features of collagen that have been reported by several studies and have potential clinical applications in the diagnosis of diseases[38,39]. Gabor wavelet transformation

features are also textural features that are used to reflect the spatial relationship of collagen in different scales and orientations after image convolution[40]. In our previous studies, we extracted four types of the above-mentioned features to evaluate liver fibrosis using multiphoton imaging[22,41]. Based on these results, we established the collagen signature from four types of collagen features.

LASSO regression aims to identify the variables and corresponding regression coefficients that lead to a model that minimizes the prediction error from high-dimensional data. In a practical sense, this constrains the complexity of the model. Additionally, LASSO regression trades off potential bias in estimating individual parameters for a better expected overall prediction and focuses on the best combination among the features[42]. In this study, the LASSO regression mainly selected Gabor wavelet features as potential predictive variables, which indicates that the combination of the three selected Gabor wavelet features and the mean of cross-link density was most associated with the risk of peritoneal metastasis. We found that there were correlations among the three Gabor wavelet transformation features (Supplementary Fig. 10). Although selecting independent features for a prediction model is one of the standard methods to construct a new model, LASSO regression has also been shown to outperform the standard methods in some settings, and has been broadly used to deal with high-dimensional data[24,25,42]. The extracted collagen features were regarded as an integrity, which should be a single parameter; thus, we used LASSO regression to construct the collagen signature.

Adiposity occurs in the serosa is variably distributed and might influence the collagen matrix. Fat content is associated with individual body mass index (BMI). We found that the distribution of the collagen signature between patients with high and low BMI was similar (Supplementary Fig. 11). Competing-risk regression showed that there was no significant association between BMI and peritoneal metastasis (SHR: 1.14 95% CI: 0.69–1.89; $P = 0.61$). These results indicated that the fat content in the serosa did not affect the construction of collagen signature. However, fat should be avoided as much as possible.

In the nomogram, tumour size, tumour differentiation status and lymph node metastasis were used as categorical variables, and the collagen signature was used as a continuous variable. Although lymph node status seems to be the most significant multivariate predictor of outcome, with the highest SHR and range, the prediction of the risk of peritoneal metastasis was always contributed by these four factors. For example, for a patient with a median collagen signature of 0.047 and a tumour size less than 4 cm with poor differentiation, the 3-year probability of peritoneal metastasis would be approximately 11% with no lymph node metastasis. If the N stage was N3a, the risk would increase to approximately 31%. Furthermore, the risk would be 40% if the N stage advanced to N3b. The AUROC would be reduced from 0.807 (nomogram based on the collagen signature) to 0.720 (lymph node metastasis alone) by removal of other variables (Supplementary Fig. 12). Other variables that were significantly associated with peritoneal metastasis will also be considered for inclusion in the prediction model in the future. Because tumour size, tumour differentiation status and lymph node metastasis are routinely evaluated in the clinic, and the collagen signature can be automatically quantified after multiphoton imaging, the risk of peritoneal metastasis after radical surgery in GC with serosal invasion can be estimated conveniently.

A well-designed prediction model could facilitate communication between physicians and patients and identify the genuine high-risk patients. The aim of precision medicine is to avoid overtreatment or undertreatment in the clinic and facilitate tailored decision-making. We envision that the nomogram will facilitate personalized medicine in GC with serosal invasion. Herein, with the assistance of the nomogram, we would like to recommend IPC for patients with a high risk of peritoneal metastasis to improve survival, and to reduce or even withhold IPC for patients with a low risk of peritoneal metastasis to decrease the risks of complications and additional financial burden.

Substantial efforts have been made for the purpose of early identification of peritoneal metastasis in patients with GC. Dong et al.[43] developed a radiomics nomogram based on the radiomics signature of the primary tumour and peritoneal region from abdominal CT to predict occult peritoneal metastasis preoperatively. Nevertheless, this research focused on the selection of patients with peritoneal metastasis to avoid unnecessary surgical procedures[43]. Kanda et al.[44–46] discovered a series of biomarkers, such as synaptotagmin XIII, synaptotagmin VIII and troponin I2, to predict peritoneal metastasis via a transcriptome analysis. However, the transcriptome data were obtained from only 16 patients. Moreover, a combined analysis, rather than an individual analysis, of a series of biomarkers as a single signature was more powerful at improving clinical management[24,25]. In this case, it is more promising for clinical applications to systematically analyze significantly expressed proteins.

However, the present study has some limitations. First, the nomogram was developed and externally validated based on two retrospective cohorts from two medical institutions; therefore, potential bias was inevitable. A prospective and multicentre trial is required to validate the performance of the nomogram. Second, the underlying mechanism of the predictive value of the collagen signature was not very clear; thus, further investigations are needed to better understand the role of the collagen signature for predicting peritoneal metastasis in GC with serosal invasion.

In conclusion, we determined that the collagen signature in the tumour microenvironment of the gastric serosa was associated with peritoneal metastasis in GC with serosal invasion. Furthermore, a competing-risk nomogram could distinguish a genuine high risk of peritoneal metastasis in GC with serosal invasion after radical surgery.

## Methods

This study was approved by the Institutional Review Board at Nanfang Hospital of Southern Medical University and the Fujian Provincial Cancer Hospital of Fujian Medical University. All procedures performed in this study involving human participants were in accordance with the Declaration of Helsinki. Written informed consent was obtained from all participants at the time of surgery.

**Study population**. This study enrolled two independent cohorts of patients diagnosed with GC with serosal invasion after radical surgery. The training cohort included 198 consecutive patients and was obtained from the Nanfang Hospital of Southern Medical University between July 1, 2011, and July 31, 2014. The inclusion criteria were patients who underwent radical gastrectomy with negative peritoneal lavage cytology and with histologically diagnosed GC with serosal invasion and patients with available clinicopathological data and a complete 3-year postoperative follow-up. We excluded patients treated with neoadjuvant radiotherapy, neoadjuvant chemotherapy or neoadjuvant chemoradiotherapy. The validation cohort comprising 145 consecutive patients was obtained from the Fujian Provincial Cancer Hospital of Fujian Medical University between July 1, 2008, and March 31, 2011, with the same criteria.

Baseline information was recorded for each patient, including age, sex, BMI, carcinoembryonic antigen (CEA) level, carbohydrate antigen 19-9 (CA19-9) level, tumour location, tumour size, Lauren classification, tumour differentiation status, lymph node metastasis (N stage), postoperative chemotherapy and follow-up data. Routine adjuvant chemotherapy was initiated after surgery if the patients' physical conditions were available according to the National Comprehensive Cancer Network guidelines. The diagnosis of peritoneal metastasis was determined by abdominal ultrasonography, computed tomography (CT) or positron emission tomography (PET)-CT, clinical signs, such as ascites, an intraabdominal mass, and even reoperation, during follow-up.

**Region of interest selection**. The formalin-fixed paraffin-embedded samples of each patient were used to determine the regions of interest for multiphoton imaging. All samples were sectioned at 5-μm thickness and processed for H&E staining. Two independent pathologists who were blinded to clinical characteristics and prognosis reassessed the invasive region of the gastric serosa using a microscope. When the two pathologists had different opinions, a third pathologist was consulted, and they discussed together to make a decision. Finally, five regions of interest with a field of view of 500 × 500 μm per sample within the invasive region of the gastric serosa were randomly selected.

**Image acquisition**. The regions of interest were imaged with a multiphoton imaging system[47]. The system contained a high-throughput scanning inverted Axiovert 200 microscope (LSM 510 META; Zeiss, Germany) equipped with a mode-locked femtosecond titanium (Ti): sapphire laser (110 fs, 76 MHz), tunable from 700 to 980 nm (Mira 900-F; Coherent, America). An acousto-optic modulator was used to control the attenuation of the laser intensity. A Plan-Apochromat 20× objective (Zeiss) was employed for focusing the excitation beam and for collecting the backward signals. The META detector collected the backward multiphoton signals from the tissue sample. The two-channel mode achieved two-photon excitation fluorescence (TPEF) and second harmonic generation (SHG), which was separated by a dichroic mirror in the detection path. One channel corresponds to a wavelength range of 430 to 708 nm to show the morphologies of the tissue components from the TPEF signals, whereas another channel covers the wavelength range from 387 to 409 nm to present the microstructures of the tissue components from the SHG signals. The excitation wavelength (λex) used in this study was 800 nm. Imaging acquisition was performed on another unstained serial section and compared with the H&E staining for histological assessment.

**Collagen feature extraction**. The extraction of collagen features was performed automatically via MATLAB 2015b (Mathworks, Natick, MA, USA)[41]. The extracted collagen features were summarized in Supplementary Table 7. Four types of collagen features were extracted in this study, including morphological features, histogram-based features, GLCM-based features and Gabor wavelet transform features. For morphological features, the SHG image was first segmented into collagen pixels and background pixels using the Gaussian mixture model method[48]. The binary collagen mask image was then processed using a fibre network extraction algorithm[49] to trace each collagen fibre in the image and to identify cross-link points, which are defined as connecting points between two or more fibres. Moreover, we quantified an orientation index to reflect the collagen alignment based on Fourier transform spectra[50]. For histogram-based features, a histogram-based approach was used. The mean, variation, skewness, kurtosis, energy and entropy were calculated from the histogram of the SHG pixel intensity distribution. We also included 80 GLCM-based texture features[51]. The contrast, correlation, energy and homogeneity were calculated from the GLCM with five different displacements of pixels at 1, 2, 3, 4 and 5 and four different directions at 0, 45, 90 and 135 degrees. In addition, forty-eight Gabor wavelet transform features were included for analysis[52]. To calculate the Gabor wavelet transform features, we convolved the SHG image with Gabor filters at four different scales and six different orientations, and the mean and variation in the magnitude of the convolution over the image at each setting were calculated. The GLCM-based features provide a second-order statistical representation of the distribution of grey levels within a specific region of interest, which in turn provide the basis for textural analysis. GLCM is built by calculating the occurrence of a certain grey level pair $i$ next to grey level $j$ at the distance $\delta$ along the direction $\alpha$. After GLCM is obtained, the probability density function, $\mathbf{P}_{\delta, \alpha}(i,j)$, of finding certain pairs of pixel intensity $i$ and $j$ are calculated. Therefore, GLCM textural analysis considers the variation in pixel grey levels within a certain distance. Histogram-based features summarize the collagen signal intensities within the region of interest, and the inter-pixel correlation is ignored. Gabor wavelet transformation is a kind of textural analysis that reflects spatial relationship of images in different scales and orientations after convolution of images[40]. In a word, these three types of textural features were used to describe the spatial distribution of the collagen from different perspectives.

**LASSO regression analysis and collagen signature establishment**. LASSO regression was used to select the potential predictive features from all collagen features, and a multiple-feature-based collagen signature was then established. LASSO regression is an effective method for high-dimensional predictors, especially in problems wherein the number of predictors far exceeds the number of observations[53]. The method uses an L1 penalty to shrink the coefficients to zero. The penalty parameter $\lambda$, also called the tuning constant, controls the strength of the penalty. If we reduce $\lambda$ and relax the penalty, then more predictors can enter the model. In this study, five-time cross validations were used to determine the optimal value of $\lambda$. Finally, the $\lambda$ was selected via 1-standard error (SE) criteria.

**Assessment of the collagen signature with peritoneal metastasis and prognosis**. Patients were classified into high and low collagen signature subgroups

according to the threshold selected by using X-tile in the training cohort[54], and the same threshold was applied to the validation cohort. Survival analyses were conducted to assess the impacts of the collagen signature on peritoneal metastasis, DFS and OS. DFS was defined as the time from surgery to recurrence at any site, or all-cause death, whichever came first. OS was defined as the interval between surgery and death from any cause.

**Development and validation of the competing-risk nomogram**. The primary end point of the analysis was the time to peritoneal metastasis. The follow-up duration to peritoneal metastasis was calculated from the date of surgery to the date when peritoneal metastasis was diagnosed or to the last follow-up, and information about the survival status and recurrence type was also documented. Fine-Gray competing-risk regression analysis was used to identify the risk factors for peritoneal metastasis in the training cohort[55]; treating deaths, local recurrence and distant metastasis before peritoneal metastasis as competing events, the SHR with the corresponding 95% CI was acquired. Variables with $P < 0.05$ in the univariate analyses were selected for the multivariate analysis. Finally, a competing-risk nomogram was constructed. A clinicopathological model containing only clinicopathological risk factors was also constructed for comparison.

The discrimination of the nomogram was measured by the C-index and the time-dependent ROC curve[56]. The calibration was graphically assessed with a calibration curve[56]. The validation cohort was analyzed to validate the performance of the nomogram[57]. C-index and AUROC were used to compare the performance between the nomogram based on the collagen signature and the clinicopathological model. The C-indexes and AUROCs of the two models were compared using Mann–Whitney $U$ test and Delong test, respectively.

**Clinical usefulness**. A decision curve analysis was performed to determine the clinical usefulness of the nomogram by calculating the net benefits at different threshold probabilities[58]. Decision curve analysis is a novel tool for assessing the potential population impact of adopting a risk prediction instrument into clinical practice, and was initially introduced by Vickers and Elkin in 2006[59]. The context for decision curve analysis is a situation in which individuals' risks for an undesirable outcome are assessed, and individuals with sufficiently high risk are recommended for some intervention or treatment. Decision curve analysis provides a net benefit, which is calculated by

$$\text{Net benefit} = \text{true positive rate} - \text{false positive rate} \times \frac{P_t}{1 - P_t}, \quad (1)$$

where $P_t$ is the threshold probability at which the expected benefit of treatment is equal to the expected benefit of avoiding treatment.

The maximum Youden index of the 3-year time-independent ROC curve of the nomogram in the training cohort was selected as the optimal cutoff value. Then, all 343 patients were divided into the high-risk and low-risk groups. The sensitivity, specificity, accuracy, PPV and NPV were calculated to evaluate the prediction performance of the nomogram.

**Statistical analysis**. Continuous variables, where appropriate, were compared by independent samples, unpaired, 2-sided $t$-test or the Mann–Whitney $U$ test. Categorical variables were compared by the $\chi^2$ test or Fisher's exact test. The Kaplan–Meier method and log-rank test were used to estimate DFS and OS, and a Cox proportional hazard regression was conducted to compute the HR. The cumulative incidence function was employed to show the cumulative peritoneal metastasis rate, and differences between the subgroups were compared using Gray's test. All statistical analyses were performed using R software (version 3.4.2) and SPSS software (version 19.0). LASSO regression was performed using the "glmnet" package. Fine-Gray competing-risk regression analysis and nomogram development were performed by the "cmprsk", "rms" and "mstate" packages. Assessment of the performance and validation of the nomogram were conducted using "pec" and "riskRegression" packages. ROC curves were plotted using "pROC" package. Decision curve analysis was performed with the function of "stdca.R". The "survminer" package was used for computing survival analyses. A two-sided $P < 0.05$ was considered statistically significant.

**Reporting summary**. Further information on research design is available in the Nature Research Reporting Summary linked to this article.

## Data availability

For ethical reasons the multiphoton images are not publicly available, but are available from the corresponding authors upon reasonable request. The remaining data are available within the article, supplementary information or available from the corresponding authors upon request. Source data are provided with this paper.

## Code availability

Associated codes used for data processing and analysis are publicly available from the GitHub using the following web link (Supplementary Software 1)[60]: https://github.com/Dexin-Chen/Peritoneal_Metastasis.

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

## Acknowledgements

This work was supported by grants from the National Natural Science Foundation of China (81773117 and 81771881), the State's Key Project of Research and Development Plan (2017YFC0108300, 2017YFC0108302 and 2019YFE0113700), the Guangdong Provincial Key Laboratory of Precision Medicine for Gastrointestinal Cancer (2020B121201004), the China Postdoctoral Science Foundation (2020M682789), the Special Fund for Guangdong Province Public Research and Capacity Building (2014B020215002), the Natural Science Foundation of Guangdong Province (2015A030308006), the Natural Science Foundation of Fujian Province (2018J07004), the Joint Funds of Fujian Provincial Health and Education Research (2019-WJ-21), the Science and Technology Program of Fujian Province (2018Y2003, 2019L3018 and 2019YZ016006), the Guangzhou Industry University Research Cooperative Innovation Major Project (201704020062), the Clinical Research Startup Program of Southern Medical University by High-level University Construction Funding of Guangdong Provincial Department of Education (LC2016PY010), the Scientific Research Foundation for High-Level Talents in Nanfang Hospital of Southern Medical University (201404280056), the Clinical Research Project of Nanfang Hospital (2018CR034, 2020CR001, and 2020CR011), the President Funding of Nanfang Hospital (2019Z023), and the Training Program for Undergraduate Innovation and Entrepreneurship (201912121008, 202012121091 and 202012121277).

## Author contributions

G.L., G.C., S.Z. and J.Y. conceived, designed and supervised the study; D.C., Z.L., W.L., M.F., W.J., G.W., F.C., J.L., H.C. and X.D. collected and assembled the data; D.C., Z.L., W.L., M.F., G.C., S.Z. and J.Y. were responsible for data analysis and interpretation; D.C., M.F., S.X., G.L., S.Z. and J.Y. wrote the manuscript with contributions from all other authors.

## Competing interests

The authors declare no competing interests.
