## [Peer Review File · Nature Communications]

Reviewers' comments:

Reviewer #1 (Remarks to the Author):

In this analysis, the authors use multiphoton imaging to quantify the extent of collagen alterations in the tumor microenvironment, to determine an association with peritoneal metastases. The least absolute shrinkage and selection operator (LASSO) regression was used to predict peritoneal metastases based on the collagenomics signature and clinicopathologic risk factors.

198 patients were analyzed in the training cohort, and 115 patients in the validation cohort. A significantly higher peritoneal metastasis rate was found in the high collagen signature patients compared to the low collagen signature. The investigators went on to incorporate tumor size, differentiation, and lymph node status into a competing-risk nomogram. The nomogram resulted in a good average concordance index. Clinical usefulness was considered achieved based on comparison to treat all or treat none strategies.

The authors are to be congratulated for investigating an area of need, in that the peritoneum is the most common site of recurrence after potentially curative resection of gastric cancer. The cohorts demonstrate significant differences in OS and DFS, when stratified according to low and high signature.

Why was size > 4 cm and not T stage utilized in the model?

The authors envision that the nomogram will facilitate personalized medicine. However, enthusiasm for this nomogram is tempered by the lack of clear cut-off level where an adjuvant therapy would not be indicated. Even in the low collagenomics signature group, the peritoneal metastasis rate is considerable.

It is unclear if this signature provides improved prediction of peritoneal metastases compared to other easily achieved clinical variables.

Reviewer #2 (Remarks to the Author):

I read with interest this study that investigates a collagen signature derived from multiphoton analysis of collagen structure of tissue that has been resected from patients with advanced gastric cancer. The authors have created a collagen-based nomogram to predict peritoneal metastasis for tumours that have serosal involvement and conclude that the addition of the collagen signature improves the ability to predict peritoneal deposits.

There are a number of clinical aspects of this study that need clarification and warrant assessment given the biology of gastric cancer differs by anatomical location and by histological type.

It is notable that the training set and validation cohorts are different clinically. The authors need to explain whether this affects the analysis they have conducted. Particularly with respect to location of the primary tumour, tumour size and lymph node status.

I am assuming all tumours involved the serosa but convention warrants a T stage should be recorded for all cancers that may have a bearing on the outcome. Were there subserosal vs serosal invasion and was the lesion a T4 lesion vs a T3 lesion?

There is considerable interest in Lauren classification of GC, particularly when Diffuse GC has an infiltrative phenotype and has poor outcome. It would be beneficial to stratify on the basis of this histological grouping given Diffuse histology at the serosa will have high propensity to metastasise to peritoneum. Indeed I am surprised multivariate analysis did not find a significant difference in undifferentiated GC which will comprise many of these. Could analysis incorporate Lauren classification as an independent variable?

Why did the authors choose Cardia as a reference for location? Cardia are less common in Chinese population and usually they are poorly prognostic and usually require more radical operations that will influence clinical outcomes.

Lymph node status seems to be the most significant multivariate predictor of outcome, with highest SHR and range. How do the authors propose to use this assay in clinical prediction? Will it be in combination with other variables and how is prediction of outcome affected by removal of clinical variables?

I have a technical question regarding the multiphoton assay to assess collagen. Was this collagen only in the serosa or was collagen matrix evaluated for the entire section? Would a negative control include serosa where there is no invasion of tumour cells? Formalin fixation tends to alter human tissue, while I assume all cases were treated equally, I am wondering about the technique and whether fixation (and possibly degree of fixation) may effect the collagen signature? For instance, would a case that was fixed overnight compared to one fixed over a few days, prior to paraffin embedding, influence the multiphoton imaging?

The regions of selection become important also. I note in Figure 1 there is one region of selection which has adipose tissue in the H&E. Adiposity does occur at the serosa and this will be variably distributed and may influence collagen matrix. How does BMI or fat content in the serosa of individuals affect the assay?

In usual circumstances when analysing a predictive assay you would create a model using the training cohort which then defines set thresholds and then use those thresholds in an independent validation cohort. It is not clear to me whether the threshold was set by the training cohort and then tested on the validation cohort. It appears the validation cohort was used as another independent cohort with different distribution and different threshold for collagenomics signature. Can the authors explain?

I am not sure the ROC in Supp Fig 6 need to have all three time points. I would have thought 3 year

cumulative outcome would be enough. Most relapses will occur within a 3 year timeframe if they are going to happen. I do not think they are significantly different.

When considering this as a diagnostic test it would be valuable to have an indication of the sensitivity and specificity as well as positive and negative predictive values to reassure clinicians they are selecting appropriate population for interventions.

My last point is a philosophical one. What do the authors propose will be the difference in management given their nomogram? Will they expect cases with N3 disease and serosal involvement to have different treatment based on their nomogram result because they are predicting less peritoneal involvement? I can understand they may advocate more intense peritoneal treatment for collagen signature positive patients, but would they advocate withholding treatment in advanced disease on the basis of the nomogram result?

Reviewer #3 (Remarks to the Author):

Although the manuscript presents a good amount of data with a large patient cohort, the current manuscript is currently not technically sound. Important details are missing. Some content is biologically incorrect.

1) The term ,collagenomics' is misleading. It implies to investigate the 'collagenome' - meaning all types and varieties of collagens. However, by this technique, only a very limited number of collagens are accessible by their endogenous signal. The authors should consider revising this term.

2) The authors talk about 'high-throughput quantitative collagen features'. Is there any structural/molecular assignment which can be linked to the different features? Especially the Gabor wavelet transformation features, which majorly influence the calculation of the 'collagenomics signature' is highly vague.

3) Is there any explanation why for the establishment of the collagenomics signature from the 4 types of collagen features (morphology, histogram, GLCM & Gabor wavelet) the LASSO regression mainly selected Gabor wavelet features as potential predictive variables (3 out of 4), while the other feature types barely seemed to influence the metastasis probability?

4) Training and validation data are always represented in 2 different figures/panels. Is this due to the variability between the two cohorts? If it was known from the clinical data already that there are differences between the tumor size, tumor location etc., it might be worth considering a pooling of all patients and randomly define a test and a training/validation set.

5) In regards to the methodology:

Were the unstained serial sections, that were used for the multiphoton imaging, treated in any way before the measurements? Was the paraffin removed, and if so how?

In the future, would the definition of the invasive region/ROI also be possible based on the MP

image or is there always an H&E section necessary to determine the ROI?

6) Fig. 1 indicates that the authors also collected TPEF signal from the tumor tissues. It is not clear why the features of these images were not included in the prediction models.

7) Does the X-tile plot (Supplemental Fig. 5) represent the data for training or the validation set? To verify the selection of the cut-off value, both plots should be shown. The presented plot does not lack any green color, red is the predominant color. What is the meaning of this? Also from Supplemental Fig. 4, it is not clear why this cut-off value was chosen. Why are the values overall lower in the validation set? Plots for training and validation cohort should have the same scale.

8) In general, the approach to extract the presented amount of image features, based on the collagen fiber structure, implements that many selected features are correlating. Only few features might be important for the predictive capacity. The authors should analyze the correlations of the features and select independent features for their prediction model.

9) Page 11, the author state that "the collagenomics signature was positively corrected with the cross-link density of collagen...". This is not surprising as the cross-link density is part of the 'Collagenomics signature calculation formula' (Appendix). However, it is not clear if this cross-link density (meaning the connections between individual collagen fibers?) is correlated to chemical crosslinks that are mostly present within a collagen fiber. The previous study from the authors (reference no 25) refers to chemical collagen crosslinks. Studies that analyze systematically the relationship of collagen network features (e.g. via SHG) and chemical crosslinking are still missing. The section in the manuscript (P. 11) needs clarification.

Reviewer #4 (Remarks to the Author):

The authors propose a multiphoton imaging-derived "collagenomics" signature that associates with a high risk of peritoneal metastasis in gastric cancer with serosal invasion. This signature is validated in an independent, external data set.

This validated "collagenomics" signature in and of itself is a novel and interesting finding, especially for those who study and treat gastric cancer. If there were further metastasis-associated multiphoton imaging-derived collagen-related findings presented across multiple cancer types, these would be of widespread interest to the greater cancer research community.

Seemingly in order to find clinically relevant use for the signature, the authors then build a nomogram that includes this signature to predict individual risk of peritoneal metastasis in GC with serosal invasion. However, there are major concerns and issues with their nomogram approach and methods.

Fundamentally, a nomogram is built to be used in the clinic. Therefore, there needs to be a well-defined clinical justification for creating one, i.e. what clinical decision will be aided by using it? And

this justification should be the overarching motivation for creating the nomogram in the first place. Instead, in the manuscript, there are only vague references to "clinical use" and "improving the prognosis" when introducing the nomogram. Even when presenting the results of decision curve analysis, the decision in question is not at all referred to.

It is not until later in the discussion that it becomes clear that there is an actual decision that could be influenced by the nomogram, namely which patients gets chosen to undergo intraperitoneal chemotherapy (IPC), which is costly and associated with a high rate of postoperative complications. This decision needs to be foregrounded as the basis for why a nomogram is justified in the first place. (As an aside, complications of IPL surgery can be incorporated into the decision curve as well. See Vickers et al 2008, DOI 10.1186/1472-6947-8-53.)

More concerning, because it is an issue that can not be addressed by reorganization of the manuscript, is the inclusion of the "collagenomics" signature into the nomogram without addressing the essential question of whether there is justification for including non-clinical variables into a nomogram at all. Does the "collagenomics" signature add on to the clinical variables already used in similar nomograms in any clinically meaningful way? If there is to be an additional variable beyond the usual clinical variables, there needs to be explicit justification for how inclusion of these new data (that require additional investment/expense) make the model perform better.

As an example of a paper that addresses both of these concerns, cited by the authors themselves, Dong et al (2019) are clear about the clinical utility of the nomogram they develop and demonstrate that a nomogram with their "radiomic" signatures performs better with respect to diagnostic accuracy than a model with clinical factors alone.

Beyond these major concerns, there are some other issues, statistical and otherwise:

1. Why dichotomize the "collagenomic" signature? Dichotomizing results in loss of information. Is there an association between the signature itself and time-to-event outcomes? If there is later a reason to dichotomize into "high" and "low" signature, be explicit about what that reason is.

2. Issues with the abstract: Multiphoton imaging should be mentioned because it is an essential part of the novelty of the finding. Also, reporting a significant association of a high collagenomics signature with a high risk of peritoneal metastasis and poor oncological outcomes with $P < 0.05$ is insufficient. The actual P-values should be shown - especially as multiple outcomes are being reported in that single sentence so that multiple testing issues are an immediate concern.

Point-by-point response letter

Reviewer #1

In this analysis, the authors use multiphoton imaging to quantify the extent of collagen alterations in the tumor microenvironment, to determine an association with peritoneal metastases. The least absolute shrinkage and selection operator (LASSO) regression was used to predict peritoneal metastases based on the collagenomics signature and clinicopathologic risk factors.

198 patients were analyzed in the training cohort, and 115 patients in the validation cohort. A significantly higher peritoneal metastasis rate was found in the high collagen signature patients compared to the low collagen signature. The investigators went on to incorporate tumor size, differentiation, and lymph node status into a competing-risk nomogram. The nomogram resulted in a good average concordance index. Clinical usefulness was considered achieved based on comparison to treat all or treat none strategies.

The authors are to be congratulated for investigating an area of need, in that the peritoneum is the most common site of recurrence after potentially curative resection of gastric cancer. The cohorts demonstrate significant differences in OS and DFS, when stratified according to low and high signature.

Response: We truly appreciate your efforts and comments on our manuscript. We have revised the manuscript according to your comments and suggestions.

Why was size >4 cm and not T stage utilized in the model?

Response: Thank you for your question. Four centimetres is the median tumour size of the enrolled patients; therefore, we divided patients into <4 cm and \geq 4 cm groups in terms of tumour size. A tumour size larger than 4 cm was found to be significantly associated with peritoneal metastasis after competing-risk regression and was thus utilized in the model. In this study, only patients with serosal invasion were included, and all enrolled patients were the same T stage. Therefore, the T stage was not utilized in the model.

The authors envision that the nomogram will facilitate personalized medicine. However, enthusiasm for this nomogram is tempered by the lack of clear cut-off level where an adjuvant therapy would not be indicated. Even in the low collagenomics signature group, the peritoneal metastasis rate is considerable.

Response: Thank you for your comments. Currently, a diagnosis of peritoneal metastasis after radical gastrectomy mainly depends on clinical signs, imaging examinations and even reoperation during the follow-up period, and a practical prediction model at the time point of surgery to predict peritoneal metastasis in GC patients with serosal invasion is still lacking. In this study, although the peritoneal metastasis rate was still considerable even in the low collagen signature group, a significantly higher peritoneal metastasis rate was observed in the high collagen signature group, which indicates that the collagen signature could identify patients who were more likely to suffer from peritoneal metastasis after radical surgery.

We chose 3 years as the time point. Then, the maximum Youden index of 0.3913 was selected as the optimal cutoff value in the training cohort, and all 343 patients were divided into high-risk and low-risk groups. Patients with predicted high risk were considered to receive hyperthermic intraperitoneal chemotherapy to reduce the risk of peritoneal metastasis. We found that the sensitivity, specificity, accuracy, negative predictive value and positive predictive value of the nomogram in the training cohort were 82.3%, 82.4%, 82.3%, 87.7% and 75.6%, respectively. In the validation cohort, the sensitivity was 81.4%, the specificity was 60.8%, the accuracy was 66.9%, the negative predictive value was 88.9% and the positive predictive value was 46.8%. In the total cohort, the sensitivity was 82.0%, the specificity was 72.4%, the accuracy was 75.8%, the negative predictive value was 88.0%, and the positive predictive value was 62.1%. We have added these results to our revised manuscript.

Page 10 Line 22 to Page 11 Line 8: “The maximum Youden index of 0.3913 of the ROC curve of the nomogram was selected as the optimal cutoff value in the training cohort, and patients were divided into high-risk and low-risk groups. We found that the sensitivity, specificity, accuracy, negative predictive value (NPV) and positive predictive value (PPV) of the nomogram in the training cohort were 82.3%, 82.4%, 82.3%, 87.7% and 75.6%, respectively. In the validation cohort, the sensitivity was 81.4%, the specificity was 60.8%, the accuracy was 66.9%, the NPV was 88.9%, and the PPV was 46.8%. In the total cohort, the sensitivity was 82.0%, the specificity was 72.4%, the accuracy was 75.8%, the NPV was 88.0%, and the PPV was 62.1% (Supplementary Table 3).”

Page 26 Line 7 to 11: “The maximum Youden index of the 3-year time-independent ROC curve of the nomogram in the training cohort was selected as the optimal cutoff value. Then, all 343 patients were divided into the high-risk and low-risk groups. The sensitivity, specificity, accuracy, PPV and NPV were calculated to evaluate the prediction performance of the nomogram.”

Supplementary Table 3. Performance evaluation of the nomogram

Index	Training cohort	Validation cohort	Total cohort
Threshold	0.3913	0.3913	0.3913
Sensitivity, %	82.3 (73.4-91.4)	81.4 (69.8-93.0)	82.0 (74.6-88.6)
Specificity, %	82.4 (74.8-89.1)	60.8 (51.0-69.6)	72.4 (66.1-78.3)
Accuracy, %	82.3 (76.8-87.4)	66.9 (59.3-74.5)	75.8 (71.4-80.2)
Negative predictive value, %	87.7 (82.5-92.8)	88.9 (82.2-95.2)	88.0 (83.7-92.2)
Positive predictive value, %	75.6 (68.5-83.3)	46.8 (40.2-54.3)	62.1 (56.7-66.8)

In addition, we have also added these results to the Discussion section.

Page 13 Line 10 to 20: “Currently, a diagnosis of peritoneal metastasis after radical surgery mainly depends on clinical signs, imaging examinations and even reoperation during the follow-up period; a practical prediction model at the time point of radical surgery to predict peritoneal metastasis in GC patients with serosal invasion is still lacking. In this study, although the peritoneal metastasis rate was considerable even in the low collagen signature group, a significantly higher peritoneal metastasis rate was found in the high collagen signature group, which indicates that the collagen signature could identify patients who were more likely to suffer from peritoneal metastasis after radical surgery. In addition, the nomogram yielded an overall sensitivity, specificity

and accuracy of 82.0%, 72.4% and 75.8%, respectively, which are adequate for reassuring clinicians when selecting an appropriate population for interventions.”

It is unclear if this signature provides improved prediction of peritoneal metastases compared to other easily achieved clinical variables.

Response: Thank you for your comments. In this study, we have constructed a collagen signature combined with a clinicopathological model. To assess whether the collagen signature could improve the prediction of peritoneal metastasis compared with other easily obtained clinical variables, we also constructed a clinicopathological model, which included tumour size, tumour differentiation status and lymph node metastasis, after univariate and multivariate competing-risk regression. Thereafter, we used the C-index and AUROC to compare the prediction performance between the collagen signature combined with the clinicopathological model and clinicopathological model alone. We found significant improvement in the C-index and AUROC in the collagen signature combined with the clinicopathological model compared with the clinicopathological model. Detailed results are provided as follows, and we have added these results to our revised manuscript.

Page 11 Line 10 to Page 12 Line 5: “

Comparison with the clinicopathological model

To evaluate the superiority of the nomogram based on the collagen signature over other easily obtained clinical variables, we excluded the collagen signature and built a clinicopathological model based on tumour size, tumour differentiation status and

lymph node metastasis (**Supplementary Table 4**). The clinicopathological model yielded average C-indexes of 0.757 (95% CI: 0.748-0.765) and 0.676 (95% CI: 0.662-0.697) in the training and validation cohorts, respectively, and the nomogram based on the collagen signature presented a more robust ability to predict peritoneal metastasis in all enrolled patients [C-index comparison: 0.779 (95% CI: 0.773-0.786) vs. 0.736 (95% CI: 0.725-0.746), $P<0.001$; $P<0.001$ and 0.016 for training and validation cohorts, respectively] (**Supplementary Table 5**). Moreover, the 3-year AUROC of the clinicopathological model was 0.787 (95% CI: 0.724-0.850) and 0.721 (95% CI: 0.635-0.807) in the training and validation cohorts, respectively (**Supplementary Figure 8**). Compared with the clinicopathological model, the nomogram based on the collagen signature also showed a significantly improved AUROC in all patients [AUROC comparison: 0.807 (95% CI: 0.760-0.855) vs. 0.762 (95% CI: 0.712-0.813), $P<0.001$; $P=0.01$ and 0.004 for the training and validation cohorts, respectively] (**Supplementary Figure 9 and Supplementary Table 6**).”

Page 25 Line 7 to 8: “A clinicopathological model containing only clinicopathological risk factors was also constructed for comparison.”

Page 25 Line 13 to 14: “C-index and AUROC were used to compare the performance between the nomogram based on the collagen signature and the clinicopathological model.”

Supplementary Table 4. Univariate and multivariate Fine-Gray regression without collagen signature

Variable	Univariate analysis		Multivariate analysis	
	SHR (95% CI)	P value	SHR (95% CI)	P value
Age	0.99 (0.98-1.02)	0.62		
Sex (Female vs. Male)	0.87 (0.53-1.43)	0.58		
BMI (≥ 24 kg/m ² vs. < 24 kg/m ²)	1.14 (0.69-1.89)	0.61		
CEA (Elevated vs. Normal)	1.08 (0.67-1.73)	0.76		
CA 19-9 (Elevated vs. Normal)	1.58 (0.98-2.55)	0.064		
Tumour location		0.56		
Antrum of the stomach	Reference	>0.99		
Body of the stomach	1.32 (0.80-2.19)	0.28		
Cardia of the stomach	1.07 (0.60-1.90)	0.82		
Tumour size (≥ 4 cm vs. < 4 cm)	3.32 (1.99-5.52)	<0.001	2.70 (1.57-4.63)	<0.001
Lauren classification (Diffuse or mixed vs. Intestinal)	1.30 (0.82-2.07)	0.27		
Differentiation status		0.001		0.038
Well + Moderate	Reference	>0.99	Reference	>0.99
Poor	2.35 (1.18-4.69)	0.015	1.75 (0.84-3.64)	0.13
Undifferentiated	3.67 (1.81-7.47)	<0.001	2.62 (1.22-5.64)	0.014
Lymph node metastasis		<0.001		0.006
N0	Reference	>0.99	Reference	>0.99
N1	3.16 (1.35-7.44)	0.008	2.79 (1.20-6.49)	0.018
N2	3.32 (1.33-8.24)	0.01	2.92 (1.22-5.64)	0.016
N3a	6.18 (2.67-14.29)	<0.001	4.13 (1.72-9.94)	0.002
N3b	9.28 (3.94-21.87)	<0.001	5.55 (2.22-13.84)	<0.001
Chemotherapy (Yes vs. No)	0.75 (0.46-1.23)	0.26		

Supplementary Table 5. C-index comparison between the two models

Model	C-index (95% CI)	P value
Training Cohort		
Nomogram	0.792 (0.784-0.798)	<0.001
Clinicopathological model	0.757 (0.748-0.765)	
Validation Cohort		
Nomogram	0.708 (0.692-0.726)	0.016
Clinicopathological model	0.676 (0.662-0.697)	
Total cohort		
Nomogram	0.779 (0.773-0.786)	<0.001
Clinicopathological model	0.736 (0.729-0.745)	

Supplementary Figure 8. The 3-year time-dependent ROC curves of the clinicopathological model in the training and validation cohorts.

Supplementary Figure 9. The ROC curves comparison between the nomogram and clinicopathological model. The blue line indicates the nomogram based on the collagen signature, and the yellow line indicates the clinicopathological model.

Supplementary Table 6. AUROC comparison between the two models

Model	AUROC (95% CI)	P value
Training Cohort		
Nomogram	0.825 (0.765-0.885)	0.01
Clinicopathological model	0.787 (0.724-0.850)	
Validation Cohort		
Nomogram	0.776 (0.699-0.853)	0.004
Clinicopathological model	0.721 (0.635-0.807)	
Total cohort		
Nomogram	0.807 (0.760-0.855)	<math>< 0.001</math>
Clinicopathological model	0.762 (0.712-0.813)	

Page 12 Line 16 to 20: “Compared to the clinicopathological model including tumour size, tumour differentiation status and lymph node metastasis, significant

improvement in the C-index and AUROC was observed in the nomogram based on the collagen signature, which indicated that the collagen signature could improve the prediction of peritoneal metastasis beyond the use of easily obtained clinical variables.”

Reviewer #2

I read with interest this study that investigates a collagen signature derived from multiphoton analysis of collagen structure of tissue that has been resected from patients with advanced gastric cancer. The authors have created a collagen-based nomogram to predict peritoneal metastasis for tumours that have serosal involvement and conclude that the addition of the collagen signature improves the ability to predict peritoneal deposits.

Response: We appreciate your effort in reviewing our manuscript.

There are a number of clinical aspects of this study that need clarification and warrant assessment given the biology of gastric cancer differs by anatomical location and by histological type.

Response: Thank you for your valuable comments. We have clarified and addressed the concerns you have raised point-by-point; please see below.

It is notable that the training set and validation cohorts are different clinically. The authors need to explain whether this affects the analysis they have conducted.

Particularly with respect to location of the primary tumour, tumour size and lymph node status.

Response: Thank you for your valuable comments. Considering the clinical differences between the training cohort and validation cohort, we speculated that they might be due to the limited sample size in the validation cohort compared to the training cohort (198 vs. 115). Thus, we have enlarged sample size and added additional 30 patients from October 1, 2010, to March 31, 2011, in the validation cohort using the same criteria. Then, we completed the multiphoton imaging of these patients. After adding these patients, no significant clinical differences were found between the training and validation cohorts. We have updated Table 1 in our revised manuscript.

Table 1. Characteristics of the patients in the training and validation cohorts

Variable	Training cohort (n=198)	Validation cohort (n=145)	P value
Age, median (IQR), years	57 (47.75 to 63.25)	57 (52 to 64)	0.11
Sex, no. (%)			
Male	137 (69.2)	98 (67.6)	0.75
Female	61 (30.8)	47 (32.4)	
BMI, no. (%)			
≥24 kg/m ²	47 (23.7)	37 (25.5)	0.71
<24 kg/m ²	151 (76.3)	108 (74.5)	
CEA, no. (%)			
Elevated	58 (29.3)	39 (26.9)	0.63
Normal	140 (70.7)	106 (73.1)	
CA 19-9, no. (%)			
Elevated	48 (24.2)	29 (20.0)	0.35
Normal	150 (75.8)	116 (80.0)	
Tumour location, no. (%)			
Cardia of the stomach	41 (20.7)	44 (30.3)	0.10
Body of the stomach	50 (25.3)	36 (24.8)	
Antrum of the stomach	107 (54.0)	65 (44.9)	
Tumour size, no. (%)			

≥4 cm	114 (57.6)	93 (64.1)	0.22
<4 cm	84 (42.4)	52 (35.9)	
Lauren classification, no. (%)			
Intestinal	71 (35.9)	61 (42.1)	0.24
Diffuse or mixed	127 (64.1)	84 (57.9)	
Differentiation status, no. (%)			
Well	11 (5.6)	6 (4.1)	
Moderate	39 (19.7)	40 (27.6)	0.33
Poor	95 (48.0)	60 (41.4)	
Undifferentiated	53 (26.7)	39 (26.9)	
Lymph node metastasis, no. (%)			
N0	52 (26.3)	29 (20.0)	
N1	45 (22.7)	25 (17.2)	
N2	35 (17.7)	40 (27.6)	0.13
N3a	38 (19.2)	33 (22.8)	
N3b	28 (14.1)	18 (12.4)	
Chemotherapy, no. (%)			
Yes	144 (72.7)	93 (64.1)	0.09
No	54 (27.3)	52 (35.9)	
Collagen signature, median (IQR)	0.047 (-0.247 to 0.397)	0.056 (-0.031 to 0.187)	0.43

I am assuming all tumours involved the serosa but convention warrants a T stage should be recorded for all cancers that may have a bearing on the outcome. Were there subserosal vs serosal invasion and was the lesion a T4 lesion vs a T3 lesion?

Response: Thank you for your questions. Peritoneal metastasis is caused by serosal invasion of the primary tumour and the subsequent shedding of malignant cells into the peritoneal cavity (Boku T, *et al*, Br J Surg, 1990, 77:436-9). The purpose of this study was to investigate the relationship between collagen changes in the tumour microenvironment of the serosa and peritoneal metastasis. We completely agree that T3 and T4 have different outcomes in GC. However, this study focused on collagen changes in the serosa and peritoneal metastasis. Therefore, only GC patients with serosal invasion were enrolled, and there was no comparison between subserosal vs.

serosal invasion.

There is considerable interest in Lauren classification of GC, particularly when Diffuse GC has an infiltrative phenotype and has poor outcome. It would be beneficial to stratify on the basis of this histological grouping given Diffuse histology at the serosa will have high propensity to metastasis to peritoneum. Indeed, I am surprised multivariate analysis did not find a significant difference in undifferentiated GC which will comprise many of these. Could analysis incorporate Lauren classification as an independent variable?

Response: Thank you for your good advice. We agree that the diffuse Lauren classification has an infiltrative phenotype and a poor outcome. However, in this study, all enrolled patients had GC with serosal invasion. When we incorporated the Lauren classification of GC in this study and conducted the competing-risk regression of the Lauren classification with peritoneal metastasis, we did not find a significant association between the Lauren classification and peritoneal metastasis (SHR: 1.30, 95% CI: 0.82-2.07; $P=0.27$). We thought that the impact of the Lauren classification on peritoneal metastasis was reduced because most GC patients with serosal invasion had the diffuse type. Similarly, patients with poorly differentiated or undifferentiated types comprised many diffuse Lauren classifications. The impact of tumour differentiation on peritoneal metastasis was also reduced in patients with serosal invasion. We have updated Table 2 in our revised manuscript.

Page 20 Line 16 to 19: “Baseline information was recorded for each patient,

including age, sex, BMI, carcinoembryonic antigen (CEA) level, carbohydrate antigen 19-9 (CA19-9) level, tumour location, tumour size, Lauren classification, tumour differentiation status, lymph node metastasis (N stage), postoperative chemotherapy and follow-up data.”

Table 2. Univariate and multivariate Fine-Gray regression in the training cohort

Variable	Univariate analysis		Multivariate analysis	
	SHR (95% CI)	P value	SHR (95% CI)	P value
Collagen signature	4.11 (2.58-6.56)	<0.001	2.49 (1.52-4.08)	<0.001
Age	0.99 (0.98-1.02)	0.62		
Sex (Female vs. Male)	0.87 (0.53-1.43)	0.58		
BMI (≥ 24 kg/m ² vs. < 24 kg/m ²)	1.14 (0.69-1.89)	0.61		
CEA (Elevated vs. Normal)	1.08 (0.67-1.73)	0.76		
CA 19-9 (Elevated vs. Normal)	1.58 (0.98-2.55)	0.064		
Tumour location		0.56		
Antrum of the stomach	Reference	>0.99		
Body of the stomach	1.32 (0.80-2.19)	0.28		
Cardia of the stomach	1.07 (0.60-1.90)	0.82		
Tumour size (≥ 4 cm vs. < 4 cm)	3.32 (1.99-5.52)	<0.001	2.35 (1.37-4.02)	0.002
Lauren classification (Diffuse or mixed vs. Intestinal)	1.30 (0.82-2.07)	0.27		
Differentiation status		0.001		0.12
Well + Moderate	Reference	>0.99	Reference	>0.99
Poor	2.35 (1.18-4.69)	0.015	1.42 (0.66-3.05)	0.37
Undifferentiated	3.67 (1.81-7.47)	<0.001	2.12 (0.96-4.72)	0.065
Lymph node metastasis		<0.001		0.060
N0	Reference	>0.99	Reference	>0.99
N1	3.16 (1.35-7.44)	0.008	2.47 (1.02-6.20)	0.044
N2	3.32 (1.33-8.24)	0.01	2.50 (1.01-6.20)	0.047
N3a	6.18 (2.67-14.29)	<0.001	3.22 (1.25-8.31)	0.016
N3b	9.28 (3.94-21.87)	<0.001	4.43 (1.65-11.94)	0.003
Chemotherapy (Yes vs. No)	0.75 (0.46-1.23)	0.26		

Why did the authors choose Cardia as a reference for location? Cardia are less common in Chinese population and usually they are poorly prognostic and usually require more radical operations that will influence clinical outcomes.

Response: Thank you for your suggestion. We have selected the antrum as the reference for the location in our revised manuscript, and the results revealed that tumour location was not associated with peritoneal metastasis. We have added this point to the revised manuscript. Please see the revised Table 2.

Lymph node status seems to be the most significant multivariate predictor of outcome, with highest SHR and range. How do the authors propose to use this assay in clinical prediction? Will it be in combination with other variables and how is prediction of outcome affected by removal of clinical variables?

Response: We deeply appreciate your comments. The prediction model contained tumour size, tumour differentiation status, lymph node status and the collagen signature. Although lymph node status seems to be the most significant multivariate predictor of outcome, with the highest SHR and range, these four factors always contributed risk predictions of peritoneal metastasis. For example, for a patient with a median collagen signature of 0.047 and a tumour size less than 4 cm with poor differentiation, the 3-year probability of peritoneal metastasis would be approximately 11% without lymph node metastasis. If the N stage was diagnosed as N3a, the risk would increase to approximately 31%. Furthermore, the risk would be 40% if the N stage advanced to N3b. Other variables that were significantly associated with peritoneal metastasis will also be considered for inclusion in the prediction model in the future.

To assess the influence of outcome prediction of lymph node status by the removal of clinical variables, we calculated the AUORC and plotted ROC curves based on a combination of lymph node metastasis and other variables. The AUROC was reduced from 0.807 (nomogram) to 0.720 (lymph node metastasis alone) by the removal of other variables. We have addressed these points in our revised manuscript.

Supplementary Figure 12. Prediction performance for peritoneal metastasis with lymph node metastasis and different combinations of variables.

Page 17 Line 9 to 20: “Although lymph node status seems to be the most significant multivariate predictor of outcome, with highest SHR and range, the prediction of the risk of peritoneal metastasis was always contributed by these four factors. For example, for a patient with a median collagen signature of 0.047 and a tumour size

less than 4 cm with poor differentiation, the 3-year probability of peritoneal metastasis would be approximately 11% without no lymph node metastasis. If the N stage was N3a, the risk would increase to approximately 31%. Furthermore, the risk would be 40% if the N stage advanced to N3b. The AUROC would be reduced from 0.807 (nomogram based on the collagen signature) to 0.720 (lymph node metastasis alone) by removal of other variables (**Supplementary Figure 12**). Other variables that were significantly associated with peritoneal metastasis will also be considered for inclusion in the prediction model in the future.”

I have a technical question regarding the multiphoton assay to assess collagen. Was this collagen only in the serosa or was collagen matrix evaluated for the entire section? Would a negative control include serosa where there is no invasion of tumour cells? Formalin fixation tends to alter human tissue, while I assume all cases were treated equally, I am wondering about the technique and whether fixation (and possibly degree of fixation) may affect the collagen signature? For instance, would a case that was fixed overnight compared to one fixed over a few days, prior to paraffin embedding, influence the multiphoton imaging?

Response: Thank you for your questions. In fact, only collagen in the serosa was evaluated, not the entire section. In this study, we focused on the local collagen changes in the tumour microenvironment of the serosa, and we presumed that there were no differences in collagen distribution and structure in normal serosa between patients with and without peritoneal metastasis; thus, a negative control of the normal

serosa was not included.

Multiphoton imaging is a label-free and noninvasive approach to detect the tissue structure and cell morphology of specimens that is comparable to H&E staining (Yan J, *et al.* Surg Endosc, 2014, 28:36-41; Yan J, *et al.* J Biomed Opt, 2012, 17:026004; Chen J, *et al.* Gastrointest Endosc, 2011, 73:802-7), and it does not affect the collagen signature. It has been reported that tissue fixation and paraffin embedding have negligible effects on collagen detection and quantification (Kakkad SM, *et al.* J Biomed Opt, 2012, 17: 116017); thus, a sample that is fixed overnight compared to one fixed over a few days prior to paraffin embedding would not influence the multiphoton imaging. We have addressed these points in the Discussion section of our revised manuscript.

Page 14 Line 20 to 22: “Moreover, multiphoton imaging is a label-free and noninvasive tool to obtain the tissue structure and cell morphology of specimens; it is comparable to hematoxylin-eosin (H&E) staining and does not affect the collagen signature^{35,36}”

Page 15 Line 2 to Page 15 Line 6: “In addition, it has been reported that tissue fixation and paraffin embedding have negligible effects on collagen detection and quantification; thus, a sample that was fixed overnight compared to one fixed over a few days, prior to paraffin embedding, would not influence multiphoton imaging³⁷.”

The regions of selection become important also. I note in Figure 1 there is one region of selection which has adipose tissue in the H&E. Adiposity does occur at the serosa and this will be variably distributed and may influence collagen matrix. How does BMI or fat content in the serosa of individuals affect the assay?

Response: Thank you for your comments and questions. To explain whether the adipose tissue in the serosa would affect the assay, we have added the BMI information of all enrolled patients. We found that there was no significant difference in the collagen signature between patients with high and low BMI in both the training and validation cohorts. In addition, competing-risk regression indicated that the BMI had no effect on peritoneal metastasis (SHR: 1.14 95% CI: 0.69-1.89; $P=0.61$). Therefore, although adiposity does occur in the serosa, we believe that the fat content in the serosa does not affect this assay. However, fat should be avoided as much as possible.

Supplementary Figure 11. Collagen signature distribution based on BMI in the (a) training and (b) validation cohorts.

Page 16 Line 20 to Page 17 Line 5: “Adiposity occurs in the serosa is variably distributed and might influence the collagen matrix. Fat content is associated with individual body mass index (BMI). We found that the distribution of the collagen signature between patients with high and low BMI was similar (**Supplementary Figure 11**). Competing-risk regression showed that there was no significant association between BMI and peritoneal metastasis (SHR: 1.14 95% CI: 0.69-1.89; $P=0.61$). These results indicated that the fat content in the serosa did not affect the construction of collagen signature. However, fat should be avoided as much as possible.”

In usual circumstances when analysing a predictive assay you would create a model using the training cohort which then defines set thresholds and then use those thresholds in an independent validation cohort. It is not clear to me whether the threshold was set by the training cohort and then tested on the validation cohort. It appears the validation cohort was used as another independent cohort with different distribution and different threshold for collagenomics signature. Can the authors explain?

Response: Thank you for your comments. The threshold of 0.2 of the collagen signature was set by the training cohort using X-tile. Then, the same threshold was applied to the validation cohort. We have provided this statement in the Methods section of our previous manuscript.

Page 24 Line 14 to 16: “Patients were classified into high and low collagen signature

subgroups according to the threshold selected by using X-tile in the training cohort⁵⁴, and the same threshold was applied to the validation cohort.”

The number of patients was different between the training and validation cohorts, which might result in the misunderstanding that the threshold of collagen signature in the validation cohort was different from that in the training cohort. The distribution of the collagen signature between the two cohorts was similar ($P=0.43$). In fact, the y-axis in the Supplementary Figure 4 also revealed that the threshold in the training cohort was consistent with that in the validation cohort. We have changed the scale of the y-axis in Supplementary Figure 4 in our revised manuscript.

Supplementary Figure 4. Distribution of the collagen signature in the training and validation cohorts. **(a)** The distribution of the collagen signature in the training cohort, with 70 patients classified into the high collagen signature group and 118 patients classified into the low collagen signature group based on a cutoff value of 0.2. **(b)** The distribution of the collagen signature in the validation cohort, with 31 patients classified into the high collagen signature group and 114 patients classified into the low collagen signature group based on a cutoff value of 0.2.

I am not sure the ROC in Supp Fig 6 need to have all three time points. I would have thought 3-year cumulative outcome would be enough. Most relapses will occur within a 3-year timeframe if they are going to happen. I do not think they are significantly different.

Response: Thank you for your suggestion. The reason we showed the ROC curve at three time points was to present the discrimination ability of the prediction model at different time points. We agree that there was no significant difference among the different time points, and a 3-year cumulative outcome would be sufficient. Therefore, we have removed the ROC curves at 1 and 2 years in our revised manuscript, and the statement in the corresponding Results section was also revised.

Supplementary Figure 6. The 3-year time-dependent ROC curves of the nomogram in training and validation cohorts.

Page 10 Line 4 to 7: “The time-dependent receiver operating characteristic (ROC) curve of the nomogram to predict peritoneal metastasis at 3 years in the training

cohort is presented in **Supplementary Figure 6a**, with an area under the receiver operating characteristic curve (AUROC) of 0.825 (95% CI: 0.765-0.885).”

Page 10 Line 10 to 11: “In the validation cohort, the AUROC at 3 years was 0.776 (95% CI: 0.699-0.853) (**Supplementary Figure 6b**).”

When considering this as a diagnostic test it would be valuable to have an indication of the sensitivity and specificity as well as positive and negative predictive values to reassure clinicians they are selecting appropriate population for interventions.

Response: Thank you for your useful comments. We chose the 3 years as the time point. Then, the maximum Youden index of 0.3913 of the ROC curve was selected as the optimal cutoff value in the training cohort, and all 343 patients were divided into high-risk and low-risk groups. Patients with predicted high risk were considered to receive hyperthermic IPC to reduce the risk of peritoneal metastasis. We found that the sensitivity, specificity, accuracy, negative predictive value and positive predictive value of the nomogram in the training cohort were 82.3%, 82.4%, 82.3%, 87.7% and 75.6%, respectively. In the validation cohort, the sensitivity was 81.4%, the specificity was 60.8%, the accuracy was 66.9%, the negative predictive value was 88.9% and the positive predictive value was 46.8%. In the total cohort, a sensitivity of 82.0%, a specificity of 72.4%, an accuracy of 75.8%, a negative predictive value of 88.0% and a positive predictive value of 62.1% were detected. We have added these results in our revised manuscript.

Page 10 Line 22 to Page 11 to 8: “The maximum Youden index of 0.3913 of the

ROC curve of the nomogram was selected as the optimal cutoff value in the training cohort, and patients were divided into high-risk and low-risk groups. We found that the sensitivity, specificity, accuracy, negative predictive value (NPV) and positive predictive value (PPV) of the nomogram in the training cohort were 82.3%, 82.4%, 82.3%, 87.7% and 75.6%, respectively. In the validation cohort, the sensitivity was 81.4%, the specificity was 60.8%, the accuracy was 66.9%, the NPV was 88.9%, and the PPV was 46.8%. In the total cohort, the sensitivity was 82.0%, the specificity was 72.4%, the accuracy was 75.8%, the NPV was 88.0%, and the PPV was 62.1% (Supplementary Table 3).”

Page 26 Line 7 to 11: “The maximum Youden index of the 3-year time-independent ROC curve of the nomogram in the training cohort was selected as the optimal cutoff value. Then, all 343 patients were divided into the high-risk and low-risk groups. The sensitivity, specificity, accuracy, PPV and NPV were calculated to evaluate the prediction performance of the nomogram.”

Supplementary Table 3. Performance evaluation of the nomogram

Index	Training cohort	Validation cohort	Total cohort
Threshold	0.3913	0.3913	0.3913
Sensitivity, %	82.3 (73.4-91.4)	81.4 (69.8-93.0)	82.0 (74.6-88.6)
Specificity, %	82.4 (74.8-89.1)	60.8 (51.0-69.6)	72.4 (66.1-78.3)
Accuracy, %	82.3 (76.8-87.4)	66.9 (59.3-74.5)	75.8 (71.4-80.2)
Negative predictive value, %	87.7 (82.5-92.8)	88.9 (82.2-95.2)	88.0 (83.7-92.2)
Positive predictive value, %	75.6 (68.5-83.3)	46.8 (40.2-54.3)	62.1 (56.7-66.8)

Page 13 Line 17 to 20: “In addition, the nomogram yielded an overall sensitivity, specificity and accuracy of 82.0%, 72.4% and 75.8%, respectively, which are adequate for reassuring clinicians when selecting an appropriate population for interventions.”

My last point is a philosophical one. What do the authors propose will be the difference in management given their nomogram? Will they expect cases with N3 disease and serosal involvement to have different treatment based on their nomogram result because they are predicting less peritoneal involvement? I can understand they may advocate more intense peritoneal treatment for collagen signature positive patients, but would they advocate withholding treatment in advanced disease on the basis of the nomogram result?

Response: Thank you for your fully philosophical questions. A well-designed prediction model could facilitate communication between physicians and patients and help to identify genuine high-risk patients. The aim of precision medicine is to avoid overtreatment or undertreatment in the clinic and facilitate tailored decision making. For example, if a patient with N3 disease and serosal involvement was predicted to have less postoperative peritoneal involvement based on our nomogram results, we would like to reduce the administration of postoperative hyperthermic IPC to avoid side effects such as haematologic toxicity and chemical peritonitis. We have added these statements in our revised manuscript.

Page 18 Line 4 to 11: “A well-designed prediction model could facilitate

communication between physicians and patients and identify the genuine high-risk patients. The aim of precision medicine is to avoid overtreatment or undertreatment in the clinic and facilitate tailored decision-making. We envision that the nomogram will facilitate personalized medicine in GC with serosal invasion. Herein, with the assistance of the nomogram, we would like to recommend IPC for patients with a high risk of peritoneal metastasis to improve survival, and to reduce or even withhold IPC for patients with a low risk of peritoneal metastasis to decrease the risks of complications and additional financial burden.”

Reviewer #3

Although the manuscript presents a good amount of data with a large patient cohort, the current manuscript is currently not technically sound. Important details are missing. Some content is biologically incorrect.

Response: We deeply appreciate your efforts on our manuscript. We have revised the manuscript according to your comments and suggestions.

1) The term ,collagenomics‘ is misleading. It implies to investigate the ‘collagenome’ - meaning all types and varieties of collagens. However, by this technique, only a very limited number of collagens are accessible by their endogenous signal. The authors should consider revising this term.

Response: Thank you for your good advice. We have changed the “collagenomics signature” to “collagen signature” throughout our revised manuscript to avoid misleading the readers. Details are presented in the manuscript.

2) The authors talk about ‘high-throughput quantitative collagen features’. Is there any structural/molecular assignment which can be linked to the different features? Especially the Gabor wavelet transformation features, which majorly influence the calculation of the ‘collagenomics signature’ is highly vague.

Response: Thank you for your comments. Multiphoton imaging can visualize biomolecular arrays in cells, tissues and organisms; thus, the structural/molecular assignment might be linked to the different features. In this work, the “high-throughput quantitative collagen features” means that the high-dimensional features of collagen including morphological and textural features from second harmonic generation (SHG) images could be extracted after image processing. Gabor wavelet transformation is a kind of textural analysis that is used to reflect spatial relationship of image in different scales and orientations after convolution (Grigorescu SE, *et al.* IEEE Trans Image Process, 2002, 11: 1160-7). Other than the visually apparent features such as the length of collagen or area covered, the Gabor wavelet transformation features in this study indicate the collagen distribution of the image in different degrees. In the revised manuscript, we have addressed these points.

Page 15 Line 8 to 12: “Multiphoton imaging can visualize biomolecular arrays in cells, tissues and organisms; thus, the structural or molecular assignment may be linked to different collagen features. In this work, the high-throughput quantitative collagen features obtained suggest that the high-dimensional features of collagen including morphological and textural features from multiphoton imaging could be extracted after image processing.”

Page 15 Line 18 to 21: “Gabor wavelet transformation features are also textural features that are used to reflect the spatial relationship of collagen in different scales and orientations after image convolution⁴⁰”

3) Is there any explanation why for the establishment of the collagenomics signature from the 4 types of collagen features (morphology, histogram, GLCM & Gabor wavelet) the LASSO regression mainly selected Gabor wavelet features as potential predictive variables (3 out of 4), while the other feature types barely seemed to influence the metastasis probability?

Response: Thank you for your comments. The SHG of multiphoton imaging was initially used to describe the collagen morphology for optical diagnosis of tissues. Collagen from SHG imaging was presented to describe empirical observations that were associated with particular pathological conditions. Multiphoton imaging has emerged as a useful tool for extracting quantitative collagen features in recent years. Currently, a common consensus about the selection of feature types has not yet been achieved to comprehensively quantify collagen alterations using multiphoton imaging. In our previous studies, we extracted four types of the abovementioned features to evaluate liver fibrosis (Xu S, *et al.* J Hepatol. 2014, 61:260-9; Xu S, *et al.* J Biophotonics. 2016, 9: 351-63). Similarly, in this study, we constructed the collagen signature based on these four types of collagen features.

The morphological features, such as collagen length and width, are easily understood.

Histogram and GLCM are two main types of textural features of collagen that have been reported by several studies (Mostaço-Guidolin LB, *et al.* Am J Respir Crit Care Med. 2019, 200: 431-43; Hristu R, *et al.* Biomed Opt Express. 2018, 9: 3923-36). The GLCM provides a second-order statistical representation of the distribution of grey levels within a specific ROI, which, in turn, provides the basis for textural analysis. GLCM is built by calculating the occurrence of a certain grey-level pair i next to grey level j at the distance δ along the direction α . After the GLCM is obtained, the probability density function, $P_{\delta, \alpha}(i, j)$, of finding certain pairs of pixel intensity i and j are calculated. Therefore, GLCM textural analysis considers the variation in pixel grey levels within a certain distance. Thereby, the forms, distributions and variation in the imaged objects, such as collagen, can be tracked (Golaraei A, *et al.* Biomed Opt Express. 2020, 11: 1851-63). Histogram-based features summarize the collagen signal intensities within the ROI, and the inter-pixel correlation is ignored. The three types of textural features, including Gabor wavelet transformation features, were used to describe the spatial distribution of the collagen from different perspectives. We have addressed these points in our revised manuscript.

Page 15 Line 12 to Page 16 Line 1: “To date, a common consensus about the selection of collagen feature types has not yet been achieved to comprehensively quantify collagen alterations based on multiphoton imaging. The morphological features, such as collagen length and width, are easily understood. Histogram- and grey-level co-occurrence matrix (GLCM)-based features are two main types of textural features of collagen that have been reported by several studies and have

potential clinical applications in the diagnosis of diseases^{38,39}. Gabor wavelet transformation features are also textural features that are used to reflect the spatial relationship of collagen in different scales and orientations after image convolution⁴⁰. In our previous studies, we extracted four types of the abovementioned features to evaluate liver fibrosis using multiphoton imaging^{22,41}. Based on these results, we established the collagen signature from four types of collagen features.”

Page 23 Line 10 to 22: “The GLCM-based features provide a second-order statistical representation of the distribution of grey levels within a specific region of interest, which in turn provide the basis for textural analysis. GLCM is built by calculating the occurrence of a certain grey level pair i next to grey level j at the distance δ along the direction α . After GLCM is obtained, the probability density function, $P_{\delta, \alpha}(i, j)$, of finding certain pairs of pixel intensity i and j are calculated. Therefore, GLCM textural analysis considers the variation in pixel grey levels within a certain distance. Histogram-based features summarize the collagen signal intensities within the region of interest, and the inter-pixel correlation is ignored. Gabor wavelet transformation is a kind of textural analysis that reflects spatial relationship of images in different scales and orientations after convolution of images⁴⁰. In a word, these three types of textural features were used to describe the spatial distribution of the collagen from different perspectives.”

LASSO regression aims to identify the variables and corresponding regression coefficients that lead to a model that minimizes the prediction error from

high-dimensional data. In a practical sense this constrains the complexity of the model. Additionally, the LASSO approach trades off potential bias in estimating individual parameters for a better expected overall prediction (Ranstam J, *et al.* Br J Surg. 2018, 105: 1348. DOI: 10.1002/bjs.10895). In this study, LASSO regression mainly selected Gabor wavelet features as potential predictive variables, which indicates that the combination of the three selected Gabor wavelet features and the mean cross-link density was most associated with the risk of peritoneal metastasis. In the revised manuscript, we have added these explanations.

Page 16 line 3 to 11: “LASSO regression aims to identify the variables and corresponding regression coefficients that lead to a model that minimizes the prediction error from high-dimensional data. In a practical sense, this constrains the complexity of the model. Additionally, LASSO regression trades off potential bias in estimating individual parameters for a better expected overall prediction and focuses on the best combination among the features⁴². In this study, the LASSO regression mainly selected Gabor wavelet features as potential predictive variables, which indicates that the combination of the three selected Gabor wavelet features and the mean of cross-link density was most associated with the risk of peritoneal metastasis.”

4) Training and validation data are always represented in 2 different figures/panels. Is this due to the variability between the two cohorts? If it was known from the clinical data already that there are differences between the tumor size, tumor location etc., it might be worth considering a pooling of all patients and randomly define a test and a

training/validation set.

Response: Thank you for your good questions. The reason we represented the training and validation cohorts in 2 different figures/panels was not due to the variability between the two cohorts. We wanted to reveal that the performance of the prediction model developed in the training cohort could be validated in the validation cohort.

In addition, considering the clinical differences between the training cohort and validation cohort, we speculate that it might be due to the limited sample size in the validation cohort compared to the training cohort (198 vs. 115). Thus, we have enlarged the sample size and added additional 30 patients from October 1, 2010, to March 31, 2011, in the validation cohort using the same criteria. Then, we completed multiphoton imaging of these patients. After adding these patients, there were not significant clinical differences between the training and validation cohort. We have updated the new Table 1 in our revised manuscript.

Table 1. Characteristics of the patients in the training and validation cohorts

Variable	Training cohort (n=198)	Validation cohort (n=145)	P value
Age, median (IQR), years	57 (47.75 to 63.25)	57 (52 to 64)	0.11
Sex, no. (%)			
Male	137 (69.2)	98 (67.6)	0.75
Female	61 (30.8)	47 (32.4)	
BMI, no. (%)			
≥ 24 kg/m ²	47 (23.7)	37 (25.5)	0.71
< 24 kg/m ²	151 (76.3)	108 (74.5)	
CEA, no. (%)			
Elevated	58 (29.3)	39 (26.9)	0.63
Normal	140 (70.7)	106 (73.1)	
CA 19-9, no. (%)			
Elevated	48 (24.2)	29 (20.0)	0.35

Normal	150 (75.8)	116 (80.0)	
Tumour location, no. (%)			
Cardia of the stomach	41 (20.7)	44 (30.3)	
Body of the stomach	50 (25.3)	36 (24.8)	0.10
Antrum of the stomach	107 (54.0)	65 (44.9)	
Tumour size, no. (%)			
≥4 cm	114 (57.6)	93 (64.1)	0.22
<4 cm	84 (42.4)	52 (35.9)	
Lauren classification, no. (%)			
Intestinal	71 (35.9)	61 (42.1)	0.24
Diffuse or mixed	127 (64.1)	84 (57.9)	
Differentiation status, no. (%)			
Well	11 (5.6)	6 (4.1)	
Moderate	39 (19.7)	40 (27.6)	0.33
Poor	95 (48.0)	60 (41.4)	
Undifferentiated	53 (26.7)	39 (26.9)	
Lymph node metastasis, no. (%)			
N0	52 (26.3)	29 (20.0)	
N1	45 (22.7)	25 (17.2)	
N2	35 (17.7)	40 (27.6)	0.13
N3a	38 (19.2)	33 (22.8)	
N3b	28 (14.1)	18 (12.4)	
Chemotherapy, no. (%)			
Yes	144 (72.7)	93 (64.1)	0.09
No	54 (27.3)	52 (35.9)	
Collagen signature, median (IQR)	0.047 (-0.247 to 0.397)	0.056 (-0.031 to 0.187)	0.43

5) In regards to the methodology:

Were the unstained serial sections, that were used for the multiphoton imaging, treated in any way before the measurements? Was the paraffin removed, and if so how?

In the future, would the definition of the invasive region/ROI also be possible based on the MP image or is there always an H&E section necessary to determine the ROI?

Response: Thank you for your questions. No treatment was performed on the unstained serial sections before the measurements, and the paraffin did not need to be removed, according to previous studies (Xu S, *et al.* J Hepatol. 2014, 61:260-9; Xu S,

et al. J Biophotonics. 2016, 9: 351-63).

Multiphoton imaging is a label-free tool to obtain the tissue structure and cell morphology of specimens, and it is comparable to H&E staining (Yan J, *et al. Surg Endosc*, 2014, 28:36-41; Yan J, *et al. J Biomed Opt*, 2012, 17:026004; Chen J, *et al. Gastrointest Endosc*, 2011, 73:802-7); thus, experienced pathologists can master multiphoton imaging with little training and define the ROIs based on multiphoton imaging. We have addressed these points to the Discussion section of our revised manuscript.

Page 14 Line 18 to Page 15 Line 6: “There was no treatment on the unstained serial sections before the measurements, and the paraffin did not need to be removed^{17,22} Moreover, multiphoton imaging is a label-free and noninvasive tool to obtain the tissue structure and cell morphology of specimens; it is comparable to hematoxylin-eosin (H&E) staining and does not affect the collagen signature^{35,36}; thus, experienced pathologists could master multiphoton imaging with little training, and it is possible to define regions of interest based on multiphoton imaging. In addition, it has been reported that tissue fixation and paraffin embedding have negligible effects on collagen detection and quantification; thus, a sample that was fixed overnight compared to one fixed over a few days, prior to paraffin embedding, would not influence multiphoton imaging³⁷.”

6) Fig. 1 indicates that the authors also collected TPEF signal from the tumor tissues. It is not clear why the features of these images were not included in the prediction models.

Response: Thank you for your comment. Indeed, the TPEF signal from the tumour tissues was not included in the prediction model. In this study, we focused on the influence of collagen changes on peritoneal metastasis. As a novel imaging technology, multiphoton imaging mainly contains two types of signals, including TPEF and SHG. From our previous studies, we found that combining TPEF with SHG could better reveal the tissue architecture and cell morphology of the specimens and was comparable to H&E staining (Yan J, *et al.* Surg Endosc, 2014, 28:36-41; Yan J, *et al.* J Biomed Opt, 2012, 17:026004; Chen J, *et al.* Gastrointest Endosc, 2011, 73:802-7). Thus, the TPEF signal was collected in our study to help to determine the regions of interest.

7) Does the X-tile plot (Supplemental Fig. 5) represent the data for training or the validation set? To verify the selection of the cut-off value, both plots should be shown. The presented plot does not lack any green color, red is the predominant color. What is the meaning of this? Also from Supplemental Fig. 4, it is not clear why this cut-off value was chosen. Why are the values overall lower in the validation set? Plots for training and validation cohort should have the same scale.

Response: Thank you for your questions. The X-tile plot in Supplemental Fig. 5 represents the data for training cohort. Usually, when developing and validating a

biomarker for individual prognosis, the cutoff value of the biomarker is determined in the training cohort; then, the same cutoff value is used in the validation cohort (Jiang Y, *et al.* Ann Surg, 2018, 267:504-13; Zhang JX, *et al.* Lancet Oncol, 2013, 14:1295-306; Huang Y, *et al.* Radiology, 2016, 281:947-57) to illustrate that the cutoff value is also available in the validation cohort.

For Supplementary Fig. 5, a detailed description was provided in the previous publication (Camp RL, *et al.* Clin Cancer Res, 2004, 10:7252-9). Briefly, in Supplementary Fig. 5a, the colours in the plot represent the strength of the association at each division, ranging from low (black) to high (bright red or green). Red represents the inverse association between the collagen signature and survival, indicates that the higher of the collagen signature is, the worse the survival. In contrast, green represents a positive association. Thus, the plot lacks green and is predominantly red. The *x*-axis represents all potential cutoff values from low to high (left to right) that define a low subset, whereas the *y*-axis represents cutoff values from high to low (top to bottom) that define a high subset. The optimum cutoff value of 0.2 is highlighted by the black dot on the *x*-axis. Thus, we have updated the figure legends of Supplementary Figure 5 in our revised Supplementary Information.

“Supplementary Figure 5. X-tile plots of the collagen signature with the cutoff value in the training cohort. **(a)** The colours in the plot represent the strength of the association at each division, ranging from low (black) to high (bright red or green). Red represents the inverse association between the collagen signature and survival,

whereas green represents a positive association. The x -axis represents all potential cutoff points, from low to high (left to right), that define a low subset, whereas the y -axis represents cutoff points from high to low (top to bottom) that define a high subset. The optimum cutoff point is highlighted by the black dot on the x -axis. **(b)** The cutoff value of the collagen signature and the numbers of patients in subgroups.”

In Supplementary Fig. 4, the cutoff value was chosen from the optimum cutoff value of 0.2 derived from the X-tile plot in Supplementary Fig. 5 in the training cohort. Then, the same cutoff value was used in the validation cohort. Although the collagen signature in the validation cohort seemed lower than that in the training cohort, our data showed the objective results, and the distribution of the collagen signature between the two cohorts was similar ($P=0.43$). Finally, we have updated the scale of the two cohorts in our revised Supplementary Fig. 4.

Supplementary Figure 4. Distribution of the collagen signature in the training and validation cohorts. **(a)** The distribution of the collagen signature in the training cohort, with 70 patients classified into the high collagen signature group and 118 patients classified into the low collagen signature group based on a cutoff value of 0.2. **(b)** The distribution of the collagen signature in the validation cohort, with 31 patients classified into the high collagen signature group and 114 patients classified into the low collagen signature group based on a cutoff value of 0.2.

8) In general, the approach to extract the presented amount of image features, based on the collagen fiber structure, implements that many selected features are correlating. Only few features might be important for the predictive capacity. The authors should analyze the correlations of the features and select independent features for their prediction model.

Response: Thank you for your comments. We agree that the collagen features might be correlated according to the approach used to extract image features. We analysed the correlations of the four selected features and found that correlations indeed existed among the three Gabor features.

Supplementary Figure 10. Correlation analysis between the selected features. The bottom left plots indicate the scatterplots between two features in turn. The top right values denote Pearson correlation coefficients with corresponding confidence intervals, and values closer to 1 identify a better correlation.

We agree that selecting independent features for a prediction model is one of the standard methods to construct a new model. LASSO regression has also been shown to outperform standard methods in some settings and has been broadly used to deal with high-dimensional data (Ranstam J, *et al.* Br J Surg. 2018, 105: 1348. DOI: 10.1002/bjs.10895). It trades off potential bias in estimating individual parameters for a better expected overall prediction and focuses on the best combination among the features. In this study, the extracted collagen features were regarded as an integrity, and we thought that the collagen signature was a single parameter, which was similar to age or sex. Thus, we used LASSO regression to construct the collagen signature.

Page 16 Line 11 to 18: “We found that there were correlations among the three Gabor wavelet transformation features (**Supplementary Figure 10**). Although selecting independent features for a prediction model is one of the standard methods to construct a new model, LASSO regression has also been shown to outperform the standard methods in some settings, and has been broadly used to deal with high-dimensional data^{24,25,42}. The extracted collagen features were regarded as an integrity, which should be a single parameter; thus, we used LASSO regression to construct the collagen signature.”

9) Page 11, the author state that “the collagenomics signature was positively corrected with the cross-link density of collagen....”. This is not surprising as the cross-link density is part of the ‘Collagenomics signature calculation formula’ (Appendix). However, it is not clear if this cross-link density (meaning the connections between

individual collagen fibers?) is correlated to chemical crosslinks that are mostly present within a collagen fiber. The previous study from the authors (reference no 25) refers to chemical collagen crosslinks. Studies that analyze systematically the relationship of collagen network features (e.g. via SHG) and chemical crosslinking are still missing.

The section in the manuscript (P. 11) needs clarification.

Response: Thank you for your correction. The cross-link density in our study indicated the physical connections between individual collagen fibres but not the chemical collagen crosslinks. We have clarified the statement in our revised manuscript.

Page 14 Line 7 to 13: “In this study, the cross-link density indicates the connections between individual collagen fibres (i.e. physical cross-link density). A previous study has reported that an increased chemical cross-link density of collagen heightened the stromal stiffness and stimulated the invasive properties of tumour cells³². Thus, whether there is any connection between the physical cross-link density and chemical cross-link density and how the physical cross-link density affects the biological behaviours of tumour cells needs to be further investigated.”

Reviewer #4

The authors propose a multiphoton imaging-derived "collagenomics" signature that associates with a high risk of peritoneal metastasis in gastric cancer with serosal invasion. This signature is validated in an independent, external data set.

Response: Thank you for your comments.

This validated "collagenomics" signature in and of itself is a novel and interesting finding, especially for those who study and treat gastric cancer. If there were further metastasis-associated multiphoton imaging-derived collagen-related findings presented across multiple cancer types, these would be of widespread interest to the greater cancer research community.

Response: Thank you for your comments. We truly appreciate your effort in reviewing our manuscript.

Seemingly in order to find clinically relevant use for the signature, the authors then build a nomogram that includes this signature to predict individual risk of peritoneal metastasis in GC with serosal invasion. However, there are major concerns and issues with their nomogram approach and methods.

Response: Thank you. We have addressed these concerns and issues with our nomogram approach and methods, and changes have been made to our manuscript.

Fundamentally, a nomogram is built to be used in the clinic. Therefore, there needs to

be a well-defined clinical justification for creating one, i.e. what clinical decision will be aided by using it? And this justification should be the overarching motivation for creating the nomogram in the first place. Instead, in the manuscript, there are only vague references to "clinical use" and "improving the prognosis" when introducing the nomogram. Even when presenting the results of decision curve analysis, the decision in question is not at all referred to.

It is not until later in the discussion that it becomes clear that there is an actual decision that could be influenced by the nomogram, namely which patients gets chosen to undergo intraperitoneal chemotherapy (IPC), which is costly and associated with a high rate of postoperative complications. This decision needs to be foregrounded as the basis for why a nomogram is justified in the first place. (As an aside, complications of IPL surgery can be incorporated into the decision curve as well. See Vickers et al 2008, DOI 10.1186/1472-6947-8-53.)

Response: Thank you for your valuable suggestions. We agree that a well-defined clinical justification is needed to build a nomogram, and we have addressed the clinical justification in the Introduction section. In the revised manuscript, we have clarified these points.

Page 4 Line 9 to Page 5 Line 15: “early detection of peritoneal metastasis is integral to improving the prognosis of GC patients with serosal invasion.

Currently, there are two therapies for preventing peritoneal metastasis for GC patients

with serosal invasion: extensive intraoperative peritoneal lavage (EIPL) and intraperitoneal chemotherapy (IPC)⁷. To date, the safety and efficacy of EIPL have been proven, but the long-term oncological outcomes are still unclear⁸. IPC was used to eliminate suspected malignant cells through locoregional chemotherapy. Several studies have reported that IPC is favourable for improving the oncological outcome and decreasing an incidence of peritoneal metastasis in GC with serosal invasion^{9,10}. However, a considerable number of GC patients will not suffer from peritoneal metastasis despite serosal invasion. Moreover, IPC is costly and associated with an increased rate of postoperative complications, including digestive fistula, haematologic toxicity and systemic sepsis¹¹. Thus, accurate prediction of the risk of peritoneal metastasis after radical gastrectomy is extremely important for the choice of IPC in GC with serosal invasion.

Peritoneal metastasis is difficult to predict on clinical grounds. Cytologic examination of peritoneal lavage, which has been used to assess the risk of peritoneal metastasis in GC with serosal invasion, has been reported to lack sensitivity because a large number of patients still die from peritoneal metastasis even though they have negative cytologic results¹². Some imaging modalities, including computed tomography (CT) and endoscopic ultrasonography (EUS), are common examination tools for GC; however, the accuracy of these imaging modalities for the diagnosis of peritoneal metastasis is not satisfactory¹³, and it is not until patients are suffering from peritoneal metastasis that these imaging modalities can identify the outcome. Considering the

limited performance of the clinical variables and the high complication rates of IPC, a novel biomarker is needed for the prediction of peritoneal metastasis in GC with serosal invasion after radical gastrectomy to influence decision making.”

More concerning, because it is an issue that can not be addressed by reorganization of the manuscript, is the inclusion of the "collagenomics" signature into the nomogram without addressing the essential question of whether there is justification for including non-clinical variables into a nomogram at all. Does the "collagenomics" signature add on to the clinical variables already used in similar nomograms in any clinically meaningful way? If there is to be an additional variable beyond the usual clinical variables, there needs to be explicit justification for how inclusion of these new data (that require additional investment/expense) make the model perform better.

As an example of a paper that addresses both of these concerns, cited by the authors themselves, Dong et al (2019) are clear about the clinical utility of the nomogram they develop and demonstrate that a nomogram with their "radiomic" signatures performs better with respect to diagnostic accuracy than a model with clinical factors alone.

Response: Thank you for your valuable suggestions. Currently, a diagnosis of peritoneal metastasis after radical surgery mainly depends on clinical signs, imaging examinations and even reoperation during the follow-up period; a practical prediction model at the time point of radical surgery to predict peritoneal metastasis in GC patients with serosal invasion is still lacking. To show the improved performance by

adding the collagen signature to clinically available data, we excluded the collagen signature and constructed a clinicopathological model according to the univariate and multivariate competing-risk regression, which included the tumour size, tumour differentiation status and lymph node metastasis. These results are presented in the following table. We have added this table to our revised Supplementary Information.

Supplementary Table 4. Univariate and multivariate Fine-Gray regression without collagen signature

Variable	Univariate analysis		Multivariate analysis	
	SHR (95% CI)	P value	SHR (95% CI)	P value
Age	0.99 (0.98-1.02)	0.62		
Sex (Female vs. Male)	0.87 (0.53-1.43)	0.58		
BMI (≥ 24 kg/m ² vs. < 24 kg/m ²)	1.14 (0.69-1.89)	0.61		
CEA (Elevated vs. Normal)	1.08 (0.67-1.73)	0.76		
CA 19-9 (Elevated vs. Normal)	1.58 (0.98-2.55)	0.064		
Tumour location		0.56		
Antrum of the stomach	Reference	>0.99		
Body of the stomach	1.32 (0.80-2.19)	0.28		
Cardia of the stomach	1.07 (0.60-1.90)	0.82		
Tumour size (≥ 4 cm vs. < 4 cm)	3.32 (1.99-5.52)	<0.001	2.70 (1.57-4.63)	<0.001
Lauren classification (Diffuse or mixed vs. Intestinal)	1.30 (0.82-2.07)	0.27		
Differentiation status		0.001		0.038
Well + Moderate	Reference	>0.99	Reference	>0.99
Poor	2.35 (1.18-4.69)	0.015	1.75 (0.84-3.64)	0.13
Undifferentiated	3.67 (1.81-7.47)	<0.001	2.62 (1.22-5.64)	0.014
Lymph node metastasis		<0.001		0.006
N0	Reference	>0.99	Reference	>0.99
N1	3.16 (1.35-7.44)	0.008	2.79 (1.20-6.49)	0.018
N2	3.32 (1.33-8.24)	0.01	2.92 (1.22-5.64)	0.016
N3a	6.18 (2.67-14.29)	<0.001	4.13 (1.72-9.94)	0.002
N3b	9.28 (3.94-21.87)	<0.001	5.55 (2.22-13.84)	<0.001
Chemotherapy (Yes vs. No)	0.75 (0.46-1.23)	0.26		

We compared the AUROC between the nomogram and clinicopathological model. A

significantly higher AUROC was observed in the nomogram than in the clinicopathological model for the training cohort [nomogram: 0.825 (95% CI: 0.765-0.885); clinicopathological model: 0.787 (95% CI: 0.724-0.850); $P=0.01$]. We then applied the two models to the validation cohort, and a significantly higher AUROC was also found [nomogram: 0.776 (95% CI: 0.699-0.853); clinicopathological model: 0.721 (95% CI: 0.635-0.807); $P=0.004$]. For the total cohort, the AUROC of the nomogram was 0.807 (95% CI: 0.760-0.855), which was also better than that of the clinicopathological model (0.762, 95% CI: 0.712-0.813; $P<0.001$). The C-indexes of the two models were also compared, and similar results were observed. We have added these results to the revised manuscript.

Page 11 Line 10 to Page 12 Line 5: “

Comparison with the clinicopathological model

To evaluate the superiority of the nomogram based on the collagen signature over other easily obtained clinical variables, we excluded the collagen signature and built a clinicopathological model based on tumour size, tumour differentiation status and lymph node metastasis (**Supplementary Table 4**). The clinicopathological model yielded average C-indexes of 0.757 (95% CI: 0.748-0.765) and 0.676 (95% CI: 0.662-0.697) in the training and validation cohorts, respectively, and the nomogram based on the collagen signature presented a more robust ability to predict peritoneal metastasis in all enrolled patients [C-index comparison: 0.779 (95% CI: 0.773-0.786) vs. 0.736 (95% CI: 0.725-0.746), $P<0.001$; $P<0.001$ and 0.016 for training and validation cohorts, respectively] (**Supplementary Table 5**). Moreover, the 3-year

AUROC of the clinicopathological model was 0.787 (95% CI: 0.724-0.850) and 0.721 (95% CI: 0.635-0.807) in the training and validation cohorts, respectively (**Supplementary Figure 8**). Compared with the clinicopathological model, the nomogram based on the collagen signature also showed a significantly improved AUROC in all patients [AUROC comparison: 0.807 (95% CI: 0.760-0.855) vs. 0.762 (95% CI: 0.712-0.813), $P < 0.001$; $P = 0.01$ and 0.004 for the training and validation cohorts, respectively] (**Supplementary Figure 9 and Supplementary Table 6**).”

Page 25 Line 7 to 8: “A clinicopathological model containing only clinicopathological risk factors was also constructed for comparison.”

Page 25 Line 13 to 14: “C-index and AUROC were used to compare the performance between the nomogram based on the collagen signature and the clinicopathological model.”

Supplementary Table 5. C-index comparison between the two models

Model	C-index (95% CI)	P value
Training Cohort		
Nomogram	0.792 (0.784-0.798)	<0.001
Clinicopathological model	0.757 (0.748-0.765)	
Validation Cohort		
Nomogram	0.708 (0.692-0.726)	0.016
Clinicopathological model	0.676 (0.662-0.697)	
Total cohort		
Nomogram	0.779 (0.773-0.786)	<0.001
Clinicopathological model	0.736 (0.729-0.745)	

Supplementary Figure 8. The 3-year time-dependent ROC curves of the clinicopathological model in the training and validation cohorts.

Supplementary Figure 9. The ROC curves comparison between the nomogram and clinicopathological model. The blue line indicates the nomogram based on the collagen signature, and the yellow line indicates the clinicopathological model.

Supplementary Table 6. AUROC comparison between the two models

Model	AUROC (95% CI)	P value
Training Cohort		
Nomogram	0.825 (0.765-0.885)	0.01
Clinicopathological model	0.787 (0.724-0.850)	
Validation Cohort		
Nomogram	0.776 (0.699-0.853)	0.004
Clinicopathological model	0.721 (0.635-0.807)	
Total cohort		
Nomogram	0.807 (0.760-0.855)	<0.001
Clinicopathological model	0.762 (0.712-0.813)	

Page 12 Line 16 to 20: “Compared to the clinicopathological model including tumour size, tumour differentiation status and lymph node metastasis, significant improvement in the C-index and AUROC was observed in the nomogram based on the collagen signature, which indicated that the collagen signature could improve the prediction of peritoneal metastasis beyond the use of easily obtained clinical variables.”

Beyond these major concerns, there are some other issues, statistical and otherwise:

1. Why dichotomize the "collagenomic" signature? Dichotomizing results in loss of information. Is there an association between the signature itself and time-to-event outcomes? If there is later a reason to dichotomize into "high" and "low" signature, be explicit about what that reason is.

Response: Thank you for your concerns. We agree that dichotomizing the signature could result in loss of information. When dichotomizing the signature, it is convenient

to generate the survival curves, and display the survival difference between high and low signature. From the perspective of clinicians, they prefer the categorical variables, such as sex and N stage, which are easy to identify the subgroup patients. Despite dichotomizing the collagen signature, we still regarded collagen signature as a continuous variable when construction of the nomogram (Table 2), which preserved the full information about the association between collagen signature and peritoneal metastasis.

2. Issues with the abstract: Multiphoton imaging should be mentioned because it is an essential part of the novelty of the finding. Also, reporting a significant association of a high collagenomics signature with a high risk of peritoneal metastasis and poor oncological outcomes with $P < 0.05$ is insufficient. The actual P -values should be shown - especially as multiple outcomes are being reported in that single sentence so that multiple testing issues are an immediate concern.

Response: Thank you for your helpful suggestions. We have mentioned multiphoton imaging in our revised Abstract. In addition, we have shown the actual P -values of the association between the collagen signature and peritoneal metastasis.

Page 3: “Accurate prediction of peritoneal metastasis for gastric cancer (GC) with serosal invasion is crucial in clinic. The presence of collagen in the tumour microenvironment affects the metastasis of cancer cells. Herein, we proposed a collagen signature, which was composed of multiple collagen features in the tumour microenvironment of the serosa derived from multiphoton imaging, to describe the

extent of collagen alterations. We found that a high collagen signature was significantly associated with a high risk of peritoneal metastasis ($P < 0.001$). A competing-risk nomogram including the collagen signature, tumour size, tumour differentiation status and lymph node metastasis was constructed. The nomogram demonstrated satisfactory discrimination and calibration. Thus, the collagen signature in the tumour microenvironment of the gastric serosa is associated with peritoneal metastasis in GC with serosal invasion, and the nomogram can be conveniently used to individually predict the risk of peritoneal metastasis in GC with serosal invasion after radical surgery.”

REVIEWERS' COMMENTS

Reviewer #1 (Remarks to the Author):

No further comments or edits requested

Reviewer #3 (Remarks to the Author):

The authors have addressed all issues raised. I have no additional comments.

Reviewer #4 (Remarks to the Author):

Your responsiveness to reviewer comments has greatly improved the manuscript.

My only remaining comments have to do with providing more details in the statistical methods section:

1. Please provide the statistical tests used for comparing C-indexes and AUROCs of different models.
2. Provide definitions for DFS and OS that include start and end dates.

Point-by-point response letter

Reviewer #1

No further comments or edits requested

Response: We truly appreciate your efforts on our manuscript.

Reviewer #3

The authors have addressed all issues raised. I have no additional comments.

Response: We really thank your comments and suggestions.

Reviewer #4

Your responsiveness to reviewer comments has greatly improved the manuscript.

Response: We are thankful for your helpful suggestions for improving our manuscript.

My only remaining comments have to do with providing more details in the statistical methods section:

1. Please provide the statistical tests used for comparing C-indexes and AUROCs of different models.

Response: Thank you for your comments. The C-index was regarded as a continuous variable in this study. Continuous variables with normal distribution were compared using a *t*-tests; otherwise were compared using a Mann-Whitney *U* test. The C-indexes were non-normal distribution in this study, thus the statistical test for comparing C-indexes of different models was using a Mann-Whitney *U* test.

Meanwhile, the statistical test used for comparing AUROCs of different models was Delong test. We have addressed this issue in the revised manuscript.

Page 26 Line 4 to 5: “The C-indexes and AUROCs of the two models were compared using Mann-Whitney *U* test and Delong test, respectively.”

2. Provide definitions for DFS and OS that include start and end dates.

Response: Thank you for your suggestion. We have provided the definitions of DFS and OS that include start and end dates in the revised manuscript.

Page 25 Line 3 to 6: “DFS was defined as the time from surgery to recurrence at any site, or all-cause death, whichever came first. OS was defined as the interval between surgery and death from any cause.”